# MaskCLIP++: A Mask-Based CLIP Fine-tuning Framework for Open-Vocabulary Image Segmentation

## Abstract

Open-vocabulary image segmentation has been advanced through the synergy between mask generators and vision-language models like Contrastive Language-Image Pre-training (CLIP). Previous approaches focus on generating masks while aligning mask features with text embeddings during training. In this paper, we observe that relying on generated low-quality masks can weaken the alignment of vision and language in regional representations. This motivates us to present a new fine-tuning framework, named MaskCLIP++, which uses ground-truth masks instead of generated masks to enhance the mask classification capability of CLIP. Due to the limited diversity of image segmentation datasets with mask annotations, we propose incorporating a consistency alignment constraint during fine-tuning, which alleviates categorical bias toward the fine-tuning dataset. After low-cost fine-tuning, combining with the mask generator in previous state-of-the-art mask-based open vocabulary segmentation methods, we achieve performance improvements of +1.7, +2.3, +2.1, +3.1, and +0.3 mIoU on the A-847, PC-459, A-150, PC-59, and PAS-20 datasets, respectively. Our code will be made publicly available.

## 1 Introduction

Image segmentation is one of the most extensively studied tasks in computer vision, which aims to partition an image into several regions where each region corresponds to the same object or shares consistent semantics. Traditional image segmentation models are often defined on a closed vocabulary. When new classes need to be segmented, it typically requires redefining the vocabulary set, annotating data, and retraining the model. However, the cost of annotating image segmentation on a large vocabulary is high. Therefore, the task of Open-Vocabulary Segmentation (OVS) has been proposed to perform image segmentation under any vocabulary set.

Large-scale image-text pre-training models, such as CLIP (Radford et al., 2021) and ALIGN (Jia et al., 2021), have demonstrated strong zero-shot recognition capabilities, sparking interest in transferring these abilities to image segmentation tasks. Several studies (Ding et al., 2023; Jiao et al., 2023; Yu et al., 2024; Jiao et al., 2024) have explored how to adapt mask generators (Cheng et al., 2021; 2022) and pre-trained CLIP to each other to achieve open-vocabulary segmentation at both the semantic and instance levels.

However, we argue that continuing to invest in improving the mask generation may not be cost-effective. For different CLIP architectures, even when the generated masks closely match the ground truth (GT) masks, the improvement in segmentation performance is limited to no more than 5% as shown in Figure 1(a). Conversely, for different mask generators, if CLIP can classify the generated masks as accurately as the GT class assignments, the segmentation performance improvement can exceed 40% as shown in Figure 1(b). Furthermore, Figure 1(a) shows that for the original CLIP, there remains a performance gap of 3.0%–4.7% between the generated masks and the GT masks, implying that continuing to use the generated masks during training may negatively impact the alignment quality between regional images and text. We confirm this inference in our experiments.

The above analysis indicates that GT masks can replace mask generators during fine-tuning CLIP with image segmentation data, which not only eliminates unnecessary training costs but also im-

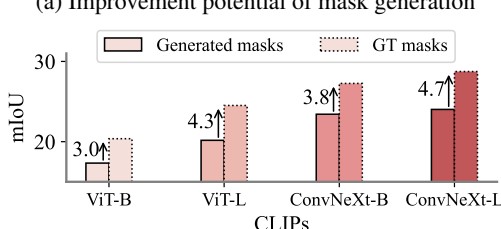
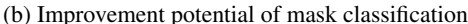
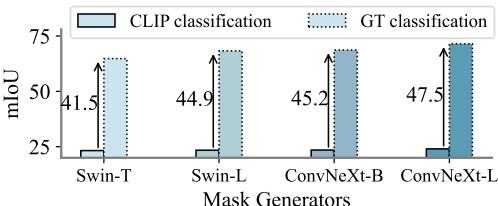

(a) Improvement potential of mask generation  (b) Improvement potential of mask classification

Figure 1: (a) Performance comparison of various original CLIP architectures using the same generated and ground truth masks. (b) Performance comparison of various mask generators with different backbones using CLIP based on ConvNeXt-L and ground truth classification. All models are trained on the COCO Panoptic dataset (Lin et al., 2014), with results reported on the ADE-150 (Zhou et al., 2017) dataset, measured by mean Intersection over Union (mIoU).

proves the alignment quality between regional images and text. However, when attempting to use image segmentation data to fine-tune CLIP, a significant challenge is to avoid overfitting to the training set. Existing approaches either integrate a frozen CLIP during inference (Xu et al., 2022b; 2023a; Yu et al., 2024) or employ a frozen teacher model for distillation (Jiao et al., 2023; 2024). The former may limit the model's performance improvement, while the latter often requires careful tuning of distillation strategies and hyperparameters. To address this, we analyze the sources of misalignment in previous methods and propose a *consistency alignment* constraint to ensure that alignment optimization preserves CLIP's original embedding space. The outcome of our explorations is a new CLIP fine-tuning framework, coined as MaskCLIP++.

Experiments show that MaskCLIP++ can significantly improve the mask classification capabilities of pretrained CLIP across multiple architectures, achieving 10.1%-16.8% mIoU improvement on ADE20K (Zhou et al., 2017). Additionally, MaskCLIP++ can be used with different mask generators to achieve open-vocabulary segmentation at the semantic- or instance-level. Compared to recent state-of-the-art methods, MaskCLIP++ offers lower training cost in terms of training time and memory usage, while achieving even better performance. We hope that the universality and efficiency of MaskCLIP++ could inspire further research in open-vocabulary image segmentation.

## 2 RELATED WORK

**Mask-based segmentation.** Mask-based segmentation is a technique that views image segmentation as the process of mask generation and mask classification, which is also known as region-based segmentation. To address instance and panoptic segmentation, He et al. (2017); Wang et al. (2020; 2021); Zhang et al. (2021) proposed various mask generators. Recently, Cheng et al. (2021) unified the semantic, panoptic and instance segmentation into a single architecture. MaskFormer (Cheng et al., 2021) uses learnable queries to interact with image features, which are then decoded as masks and class scores. Mask2Former (Cheng et al., 2022) further accelerates the convergence of MaskFormer and reduces memory usage through multi-scale decoder design, using masks from shallow decoders as priors for deeper layers, and employing sampling-based optimization. Current mask-based OVS methods need almost to retrain an open-vocabulary mask generator based on Mask2Former (Xu et al., 2023d;a; Yu et al., 2024). In contrast, our method can directly use the Mask2Former trained on a closed vocabulary during inference, reducing training cost and increasing flexibility in usage.

**Open-vocabulary segmentation.** Depending on the data used, existing methods have developed into different paradigms. The first paradigm offers many heuristic solutions based on observations of CLIP without additional training (Zhou et al., 2022; Wang et al., 2024a; Li et al., 2023; Sun et al., 2024; Lan et al., 2024a;b). The second paradigm utilizes image-text pair data (Thomee et al., 2016; Changpinyo et al., 2021) to train OVS models under *weak supervision* (Xu et al., 2022a; Luo et al., 2023; Xu et al., 2023b; Zhang et al., 2023; Mukhoti et al., 2023; Cha et al., 2023; Liu et al., 2024a). Due to the lack of precise segmentation locations, these methods suffer from poor segmentation quality. The third paradigm enhances segmentation quality by incorporating image-

mask data (Kirillov et al., 2023) and knowledge from SAM (Wang et al., 2024c; Yuan et al., 2024). However, the use of SAM not only increases inference cost but also relies on users or other detectors to provide high-quality prompts. The fourth paradigm seeks to achieve mutual adaptation between the segmentation model and CLIP through *fully supervised* training on image segmentation datasets containing image, mask, and category triplets, thereby producing an open-vocabulary segmentation model. Based on the type of the segmentation model, these methods can be categorized into mask-based approaches (Ding et al., 2022; Xu et al., 2022b; Ghiasi et al., 2022; Han et al., 2023; Liang et al., 2023; Qin et al., 2023; Jiao et al., 2023; Ding et al., 2023; Xu et al., 2023d;c;a; Yu et al., 2024; Liu et al., 2024b; Jiao et al., 2024) and pixel/patch-based ones (Li et al., 2022; Zhou et al., 2023; Li et al., 2024; Cho et al., 2024; Xie et al., 2024). Pixel-based OVS methods typically focus on semantic segmentation and cannot perform instance-level segmentation independently. Additionally, some unique approaches to achieving OVS involve image retrieval (Shin et al., 2022; Barsellotti et al., 2024), feature distillation (Chen et al., 2023; Wu et al., 2024) or model merging (Wang et al., 2024b). Some unified large models also incorporate the functionality of OVS (Zou et al., 2023; Shen et al., 2024). This paper focuses on efficiently transferring CLIP on limited segmentation datasets to improve its performance on semantic- and instance-level open-vocabulary segmentation tasks.

## 3 METHODOLOGY

### 3.1 REVISITING IMAGE-TEXT EMBEDDINGS ALIGNMENT FOR OVS

We first revisit the roles of the visual encoder (CLIP-V) and text encoder (CLIP-T) in the pretrained CLIP model. Denote the image embeddings from CLIP-V as $E_i \in \mathbb{R}^{1 \times D}$, where $D$ represents the dimension of embedding space. To create text embeddings, a vocabulary set of $K$ categories is transformed into the corresponding sentences using templates like "A photo of {}", and then encoded by CLIP-T, yielding $E_t \in \mathbb{R}^{K \times D}$. The original CLIP model maps both images and text into an aligned embedding space. In this space, image-text pairs maintain semantic alignment, which we describe as "pre-alignment" and can often be measured using cosine similarity $\langle E_i, E_t \rangle$. In open-vocabulary segmentation, preserving the pre-alignment property is crucial for adapting CLIP to zero-shot dense predictions, as it enhances the alignment of region-specific image and text embeddings.

Previous mask-based OVS methods obtain pre-aligned mask embeddings and text embeddings in two ways. As illustrated in Figure 2(a), the first way aims to train a mask generator that adapts to CLIP, with the generalization mainly derived from the original frozen parameters of CLIP (Xu et al., 2023c; Ding et al., 2023). However, attempts to align the embedding spaces of the mask generator and CLIP often fail (Xu et al., 2022b; Yu et al., 2024). As illustrated in Figure 2(b), the second way focuses on making CLIP sensitive to the generated masks during training to enhance performance and avoids overfitting through distillation (Jiao et al., 2023; 2024).

We identify two limitations of these two types of methods. First, they rely on noisy generated masks, which can degrade the quality of the region representations (as shown by the vertical comparisons in Figure 1(a)). This degradation can weaken the alignment between region-specific image and text embeddings. Second, there is a lack of an intuitive explanation and a straightforward solution for learning better alignment without severe overfitting. Moreover, freezing CLIP limits its transferability from global to local features, while distillation typically requires carefully designed strategies and incurs additional training cost.

To overcome these limitations, we propose a new training pipeline for OVS, named MaskCLIP++, which, to our knowledge, is the first method that attempts to leverage ground truth (GT) masks to obtain region features instead of relying on the mask generator during training. Furthermore, we propose the constraint named "consistency alignment" to alleviate overfitting when adding extra parameters to optimize alignment. We will give detailed descriptions in what follows.

### 3.2 A NEW FINE-TUNING FRAMEWORK FOR CLIP

In this subsection, we detail how to use our framework to fine-tune CLIP and integrate it with an off-the-shelf mask generator. The overall framework is illustrated in Figure 3.

**Extracting mask embeddings.** Disregarding architectural details, CLIP-V can be considered as consisting of two parts: feature extraction by the encoder and feature aggregation by the neck. The

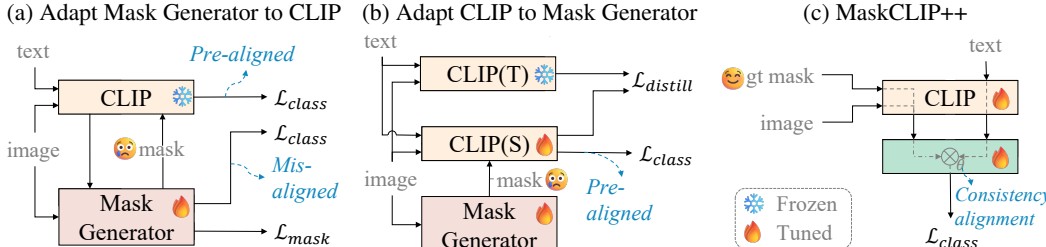

Figure 2: Comparison of training pipeline between previous mask-based OVS methods. $\mathcal{L}$ denotes the loss function. (a) adapts the mask generator to CLIP and avoids overfitting by freezing CLIP. (b) adapts CLIP to the mask generator and avoids overfitting by distillation. "T" and "S" denote the teacher and student models, respectively. Our method (c) abandons the mask generator and avoids overfitting by consistency alignment.

mask is inserted after the last feature $F_n \in \mathbb{R}^{C \times H \times W}$ of the encoder is obtained, converting the original global aggregation into an aggregation within the mask's effective region as follows:

$$E_m = \text{Proj} \sum_{i,j} \left( \phi(M)_{i,j} \cdot (F_n)_{:,i,j} \right), \tag{1}$$

where $M$ is an object mask that has been downsampled to the same size as $F_n$, $E_m \in \mathbb{R}^{1 \times D}$ is the embedding representations of this mask, $i, j$ are the spatial coordinates, and $\text{Proj}$ represents the formulation of the neck that implements channel projection. The weighting function $\phi$ is related to the original CLIP pooling structure. Specifically, if the original CLIP uses global average pooling, $\phi$ acts as an L1 normalization function to achieve mask average pooling. If the original CLIP uses attention pooling, $\phi$ can be expressed as follows:

$$\phi(M) = \text{Softmax} \left( \frac{q(T_{\text{cls}})k(T_{ij})}{\sqrt{d}} + \alpha M_{\text{bin}} \right), \tag{2}$$

where $T_{\text{cls}}$ and $T_{ij}$ represent the global and local representations sent to the attention pooling, respectively. $q, k$ are the input projection of the attention. $d$ is the hidden size of attention. The spatially expanded and binarized version of $M$ is denoted as $M_{\text{bin}}$, where any elements less than $\max(M)/2$ are set to $-\infty$.

**Parameterized similarity modeling with consistency alignment.** The pretrained CLIP mask embedding $E_m$ and the category text embedding $E_t$ are initially pre-aligned with the initial similarity map $S = \langle E_m, E_t \rangle$ being relatively coarse. Optimizing this alignment with some supervision is needed to achieve a more accurate similarity map. We refer to this process as parameterized similarity modeling (PSM).

However, when incorporating the mask generator into similarity modeling, the mask embeddings produced by the mask generator align almost exclusively with CLIP's text embeddings on the training set, resulting in the misalignment illustrated in Figure 2(a). We first analyze the cause of this overfitting to develop our similarity modeling approach.

To facilitate discussion, we simplify this approach as follows. Let $t \in \mathbb{R}^D$ denote a normalized text embedding from CLIP, and let $m_1, m_2 \in \mathbb{R}^D$ denote different mask embeddings. The initial similarity is defined as $s = m^T t$. Now, before computing the inner product across modalities, the mask embeddings are updated using a parameter $\Theta \in \mathbb{R}^{D \times D}$, resulting in a new similarity $r = (\Theta m)^T t$. Although $r$ can be optimized to be close to the target, an unintuitive phenomenon may arise: After mask embeddings are updated via gradient descent, the order of $r_1$ and $r_2$ is closer to the target, but the order of $s_1$ and $s_2$ may deviate further from the target. This phenomenon is illustrated by a toy example in Appendix Figure 5.

Consequently, the original similarity relationships between CLIP's embeddings are disrupted. Our experiments demonstrate that the additional parameter $\Theta$ fails to achieve CLIP's level of generalization when fine-tuning on limited image segmentation data, resulting in overfitting. Therefore, when attempting to parameterize similarity modeling, it is essential to ensure that the modeled similarity

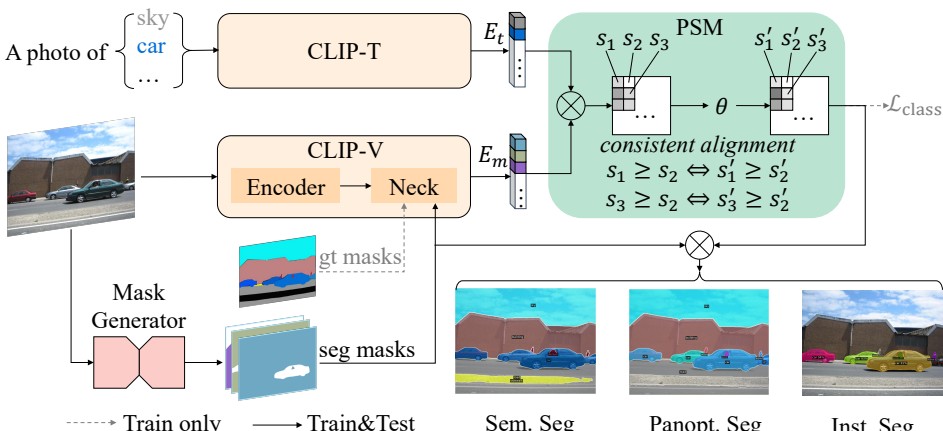

Figure 3: The detailed framework of MaskCLIP++. The PSM represents the parameterized similarity modeling, which is constrained by the consistency alignment. The mask generator is only used during inference, and can be flexibly replaced.

maintains the same order as the original similarity, so that both can be optimized toward the same target. We refer to this constraint as "consistency alignment".

To optimize under the consistency alignment constraint in most of the time, the simplest way to achieve this is to update the similarity map in a dimension that is irrelevant to any modality using linear layers. Specifically, for $Q$ masks and $K$ texts, the $Q \times K$ similarity matrix $S$ is first expanded to $Q \times K \times T$ using a linear layer of size $T$, and then it is reduced to $Q \times K$ using another linear layer of the same size.

When using the image patch embeddings $E_p$ to construct pseudo text embeddings (Jiao et al., 2024), a residual connection to the original text embeddings is necessary to ensure consistency alignment to some extent, as shown below:

$$S = \langle E_m, \mathrm{Norm}(E_t + P_t) \rangle, \quad \text{where } P_t = \mathrm{Attention}(q(E_t), k(E_p), v(E_p)) \tag{3}$$

Here, $\mathrm{Attention}$ denotes the scaled dot-product attention (Vaswani, 2017), $q, k, v$ denote the input projection functions for attention and $\mathrm{Norm}$ denotes the layer norm (Lei Ba et al., 2016). The image patch embeddings $E_p$ are equivalent to specialized mask embeddings extracted from unit masks on a per-patch basis.

**Integrating with off-the-shelf mask generators.** As the training process is entirely decoupled from the mask generator, we can flexibly employ various types of mask generators during inference to achieve semantic- or instance-level open-vocabulary segmentation. Moreover, integration with the class probabilities of the mask generator can appropriately improve the mask classification performance in the training categories. Specifically, let $\mathcal{K}$ be the training vocabularies of the mask generator. $P_c$ and $P_s$ are the class probabilities of the fine-tuned CLIP and the mask generator, respectively. The integrated class probabilities $P_\gamma$ can be expressed as follows:

$$P_\gamma^{(i)} = \begin{cases} P_s^{(i)\gamma} \cdot P_c^{(i)(1-\gamma)}, & \text{if } i \in \mathcal{K} \\ P_c^{(i)}. & \text{otherwise} \end{cases} \tag{4}$$

Compared to previous OVS methods, our MaskCLIP++ offers the following several advantages. First, we use GT masks during training, which are less noisy and more accurate, thus improving alignment quality and reducing training cost. Second, from the perspective of avoiding overfitting, the consistency alignment constraint provides more optimization space than freezing CLIP (Xu et al., 2023c; Yu et al., 2024), and does not rely on carefully designed distillation strategies (Wu et al., 2024; Jiao et al., 2024). resulting in faster convergence with a single optimization objective.

## 4 EXPERIMENTS

### 4.1 IMPLEMENTATION DETAILS

In the following experiments, we utilize various architectures of pre-trained CLIP to demonstrate the generality of our method. The CLIP models with ResNet architecture are sourced from OpenAI (Radford et al., 2021), those with ViT architecture from EVA-CLIP (Sun et al., 2023), and those with ConvNeXt architecture from OpenCLIP (Cherti et al., 2023).

By default, consistent with previous work on universal open-vocabulary semantic segmentation (Yu et al., 2024), we fine-tune our model using the COCO Panoptic training set (Lin et al., 2014), which consists of 118K images and 133 categories. The AdamW optimizer is utilized with an initial learning rate of 2e-4, decaying to 1e-5 via cosine annealing. The fine-tuning process is conducted for 20K iterations with a batch size of 4.

The size of the training image is adjusted according to the downsampling factor of the model to ensure that the final size of the feature map is $32^2$. The parameters involved in fine-tuning differ slightly across architectures. For the ConvNeXt and ResNet architectures, all parameters of CLIP-V are fine-tuned. For the ViT architecture, following Cho et al. (2024), only the positional encodings and embedding layers of CLIP-V are fine-tuned. In the ConvNeXt architecture, the fine-tuning learning rate for CLIP parameters is 1e-3 times that of other parts, while in the ResNet and ViT architectures, it is 1e-2 times.

During inference, we mainly evaluate using the ADE20K validation set (Zhou et al., 2017), which contains 2,000 images and 150 categories, along with annotations for semantic segmentation, panoptic segmentation, and instance segmentation. In ADE20K, 64 categories overlap with those in COCO, while the remaining 86 categories do not. We measure semantic segmentation performance using Intersection over Union (IoU), recording mIoU(S) for overlapping categories, mIoU(U) for non-overlapping categories, and overall mIoU. Panoptic Quality (PQ) and Average Precision (AP) are used to assess performance in panoptic segmentation and instance segmentation, respectively.

By default, we use FC-CLIP (Yu et al., 2024) trained on the COCO Panoptic (Lin et al., 2014) training set as the mask generator. For the base and large level CLIP, we employ the corresponding FC-CLIP models as mask generators [1]. The inference size of the mask generator remains consistent with FC-CLIP, set to 800. The inference size of CLIP depends on the model architecture, and the input sizes and initial segmentation performance for different architectures can be seen in Table 4.

Table 1: Importance of consistency alignment (C.A.). $\langle \cdot \rangle$ denotes cosine similarity. ✓means the consistency alignment is maintained in most of the time.

| PSM | C.A. | mIoU(S) | mIoU(U) | mIoU |
|---|---|---|---|---|
| Baseline | - | 32.7 | 16.3 | 23.3 |
| $\langle \text{Linear}(E_m), E_t \rangle$ | ✗ | 32.4 | 1.9 | 14.9 |
| $\langle E_m, \text{Linear}(E_t) \rangle$ | ✗ | 32.5 | 1.8 | 14.9 |
| $\text{Linear}\langle E_m, E_t \rangle$ | ✓ | 45.4 | 25.7 | 34.1 |
| $\langle E_m, \text{Norm}(P_t) \rangle$ | ✗ | 31.9 | 4.5 | 16.2 |
| $\langle E_m, \text{Norm}(E_t + P_t) \rangle$ | ✓ | **46.3** | **26.7** | **35.1** |

Table 2: Impact of prior quality during training. "Seg masks" is the matched generated masks. "GT boxes" is the minimum bounding rectangles of GT masks.

| Prior Type | mIoU(S) | mIoU(U) | mIoU |
|---|---|---|---|
| Pixels | 44.3 | 24.2 | 32.8 |
| Seg masks | 44.9 | 23.7 | 32.7 |
| GT boxes | 45.4 | 25.8 | 34.2 |
| GT masks | **46.3** | **26.7** | **35.1** |

### 4.2 ABLATION STUDY OF MASKCLIP++

In our ablation study, we used CLIP ConvNeXt-B along with the mask generator from FC-CLIP (Yu et al., 2024), which shares the same backbone. The direct combination of them achieved a baseline mIoU of 23.3 on ADE20K (Zhou et al., 2017).

**Importance of consistency alignment.** We use different parameterized similarity modeling methods to illustrate the importance of the consistent alignment constraint. As shown in Table 1, the attempts to use linear parameters to update mask embeddings or text embeddings violate the constraint

---

[1]Since FC-CLIP did not provide a mask generator with CLIP ConvNeXt-B as the backbone, we used their code for CLIP ConvNeXt-L to produce a base one.

Table 3: Impact of different mask generators. "OV" indicates that the mask generator utilizes a pre-trained CLIP as the backbone and employs CLIP text embeddings as a variable-length classification head, typically considered an open-vocabulary mask generator. $P_s$, $P_c$ and $P_\gamma$ are defined in equation 4. mIoU are reported. Latency (ms) is the average inference time of a single image on ADE20K, measured in an environment with Pytorch 2.3.1, cuda 11.8 and one RTX 2080Ti GPU.

| Mask Generator | | | | + MaskCLIP++ | | | | |
|---|---|---|---|---|---|---|---|---|
| Backbone | Decoder | OV | $P_s$ | $P_c$ | $P_{\gamma=0.1}$ | $P_{\gamma=0.2}$ | $P_{\gamma=0.4}$ | Latency (ms) |
| Swin-T | Mask2Former | ✗ | 15.7 | 33.9 | **34.3** | 34.2 | 33.5 | 15.5 |
| Swin-L | Mask2Former | ✗ | 16.8 | 33.9 | **34.3** | 33.9 | 33.2 | 33.6 |
| ConvNeXt-B | FC-CLIP | ✓ | 19.1 | 35.1 | 35.7 | **35.8** | 35.0 | 32.3 |
| ConvNeXt-L | FC-CLIP | ✓ | 20.8 | 34.6 | 35.4 | 35.8 | **35.8** | 39.0 |

Table 4: Improvements of various CLIP architectures. All models are fine-tuned on COCO Panoptic and evaluated on ADE20K. "Res." denotes the input resolution of CLIP during inference. For CNN-based CLIP, all parameters in CLIP-V are fine-tuned. For ViT-based CLIP, only the position embedding and input projections of the attention in the CLIP-V are fine-tuned. For RN50x16, ConvNeXt-B, and ViT-B/16, use the mask generator with ConvNeXt-B backbone. For others, use the mask generator with ConvNeXt-L backbone. Ensemble strategy is not used.

| CLIP | fine-tuned | | Input | Original CLIP | | | MaskCLIP++ | | |
|---|---|---|---|---|---|---|---|---|---|
| Arch. | Part | Params(M) | Res | mIoU | PQ | AP | mIoU | PQ | AP |
| RN50x16 | V-All | 167.3 | 512 | 18.0 | 15.7 | 9.6 | $29.6_{+11.6}$ | $22.0_{+6.3}$ | $11.2_{+1.6}$ |
| ConvNeXt-B | V-All | 90.3 | 512 | 23.3 | 19.1 | 11.6 | $35.1_{+11.8}$ | $24.5_{+5.4}$ | $13.6_{+2.0}$ |
| ConvNeXt-L | V-All | 202.7 | 640 | 25.4 | 20.7 | 14.6 | $35.6_{+10.2}$ | $26.5_{+5.8}$ | $16.7_{+2.1}$ |
| ConvNeXt-XXL | V-All | 846.5 | 640 | 26.3 | 20.3 | 13.3 | $36.4_{+10.1}$ | $27.1_{+6.8}$ | $16.6_{+3.3}$ |
| ViT-B/16 | V-Attn | 14.3 | 384 | 17.0 | 15.0 | 9.5 | $33.8_{+16.8}$ | $24.4_{+9.4}$ | $13.2_{+3.7}$ |
| ViT-L/14 | V-Attn | 50.9 | 392 | 20.3 | 17.4 | 12.4 | $36.6_{+16.3}$ | $27.3_{+9.9}$ | $17.0_{+4.6}$ |
| ViT-G/14 | V-Attn | 238.3 | 336 | 20.7 | 18.6 | 13.5 | $36.8_{+16.1}$ | $27.7_{+9.1}$ | $17.1_{+3.6}$ |

of consistency alignment during fine-tuning, resulting in a significant performance drop on unseen categories compared to the baseline (1.9% vs. 16.3% or 1.8% vs. 16.3%). The $P_t$ are weighted sums of image patch embeddings as shown in equation 3, so $\langle E_m, \text{Norm}(P_t) \rangle$ cannot have a pre-aligned initial similarity map, leading to substantial performance drops on unseen categories as well (4.5% vs. 16.3%). The PSM of $\text{Linear}\langle E_m, E_t \rangle$ adheres to the consistency alignment constraint, resulting in performance improvements on both seen and unseen categories (45.4% vs. 32.7% and 25.7% vs. 16.3%). For $\langle E_m, \text{Norm}(E_t + P_t) \rangle$, the image context information $P_t$ indirectly interacts with $E_m$. This compensates for the rigidity of mask average pooling in the CLIP ConvNeXt architecture, ultimately resulting in improved performance (35.1% vs. 34.1%). When fine-tuning CLIP models with attention pooling (ViT and ResNet), we still use the PSM of $\text{Linear}\langle E_m, E_t \rangle$. We present additional experiments in Appendix Table 11 to demonstrate the extent to evaluate how the PSMs in Table 1 preserve or disrupt the consistency alignment constraint.

**Impact of prior quality during training.** We refer to the regional information fed into CLIP during the mask embedding extraction stage as "priors" and investigate how the quality of these priors affects fine-tuning performance. We experiment with pixels, generated masks that match GT masks according to IoU, GT masks, and the GT boxes as their minimum bounding rectangles. As shown in Table 2, using GT masks can achieve the best performance. Using the generated masks as prior yields lower performance compared to the GT masks (32.7% vs. 35.1%), indicating that incorporating the mask generator during training actually hinders CLIP fine-tuning. Both GT masks and GT boxes priors outperform pixel priors, suggesting that fine-tuning CLIP from global recognition to region-level recognition is less challenging than fine-tuning it to pixel-level recognition.

**Impact of different mask generators.** Various types of mask generators can be integrated with our fine-tuned CLIP, as shown in Table 3. We use mask generators from Mask2Former (Cheng et al., 2022) and FC-CLIP (Yu et al., 2024), which are typically considered as closed-vocabulary and open-vocabulary mask generators, respectively. These mask generators are all trained on COCO

Table 5: Comparisons with previous methods on Open-Vocabulary Semantic Segmentation task.

| Method | VLM | Training dataset | A-847 | PC-459 | A-150 | PC-59 | PAS-20 |
|---|---|---|---|---|---|---|---|
| ***Non-mask-based OVS method*** | | | | | | | |
| SED (Xie et al., 2024) | CLIP ConvNeXt-B | COCO-Stuff | 11.4 | 18.6 | 31.8 | 57.7 | 94.4 |
| CAT-Seg (Cho et al., 2024) | CLIP ViT-B/16 | COCO-Stuff | 12.0 | **19.0** | 31.8 | 57.5 | 94.6 |
| ***Mask-based OVS method*** | | | | | | | |
| ZSSeg (Xu et al., 2022b) | CLIP ViT-B/16 | COCO-Stuff | 7.0 | - | 20.5 | 47.7 | - |
| OpenSeg (Ghiasi et al., 2022) | ALIGN EN-B7 | COCO-Captions + Loc. Narr. | 8.8 | 12.2 | 28.6 | 48.2 | 72.2 |
| DeOP (Han et al., 2023) | CLIP ViT-B/16 | COCO-Stuff-156 | 7.1 | 9.4 | 22.9 | 48.8 | 91.7 |
| FreeSeg (Qin et al., 2023) | CLIP ViT-B/16 | COCO-Panoptic | - | - | 24.6 | - | - |
| MAFT (Jiao et al., 2023) | CLIP ViT-B/16 | COCO-Stuff | 10.1 | 12.8 | 29.1 | 53.5 | 90.0 |
| SAN (Xu et al., 2023c) | CLIP ViT-B/16 | COCO-Stuff | 10.1 | 12.6 | 27.5 | 53.8 | 94.0 |
| SCAN (Liu et al., 2024b) | CLIP ViT-B/16 | COCO-Stuff | 10.8 | 13.2 | 30.8 | 58.4 | **97.0** |
| FC-CLIP (Yu et al., 2024) | CLIP ConvNeXt-B | COCO-Panoptic | 13.1 | 16.5 | 31.2 | 55.2 | 94.2 |
| MAFT+ (Jiao et al., 2024) | CLIP ConvNeXt-B | COCO-Stuff | 13.8 | 16.2 | 34.6 | 57.5 | 95.4 |
| MaskCLIP++ | CLIP ConvNeXt-B | COCO-Stuff | **14.5** | 18.7 | **35.4** | **59.1** | 95.8 |
| ***Non-mask-based OVS method*** | | | | | | | |
| OVSeg (Liang et al., 2023) | CLIP ViT-L/14 | COCO-Stuff | 9.0 | 12.4 | 29.6 | 55.7 | 94.5 |
| CLIPSelf (Wu et al., 2024) | CLIP ViT-L/14 | COCO-Stuff | 12.4 | - | 34.5 | 62.3 | - |
| SED (Xie et al., 2024) | CLIP ConvNeXt-L | COCO-Stuff | 13.7 | 22.1 | 35.2 | 60.6 | 96.1 |
| CAT-Seg (Cho et al., 2024) | CLIP ViT-L/14 | COCO-Stuff | 16.0 | 23.8 | 37.9 | **63.3** | 97.0 |
| ***Mask-based OVS method*** | | | | | | | |
| MaskCLIP (Ding et al., 2023) | CLIP ViT-L/14 | COCO-Panoptic | 8.2 | 10.0 | 23.7 | 45.9 | - |
| MasQCLIP (Xu et al., 2023d) | CLIP ViT-L/14 | COCO-Panoptic | 10.7 | 18.2 | 30.4 | 57.8 | - |
| ODISE (Xu et al., 2023a) | CLIP ViT-L/14 | COCO-Panoptic | 11.1 | 14.5 | 29.9 | 57.3 | - |
| SAN (Xu et al., 2023c) | CLIP ViT-L/14 | COCO-Stuff | 12.4 | 15.7 | 32.1 | 57.7 | 94.6 |
| SCAN (Liu et al., 2024b) | CLIP ViT-L/14 | COCO-Stuff | 14.0 | 16.7 | 33.5 | 59.3 | **97.2** |
| FC-CLIP (Yu et al., 2024) | CLIP ConvNeXt-L | COCO-Panoptic | 14.8 | 18.2 | 34.1 | 58.4 | 95.4 |
| MAFT+ (Jiao et al., 2024) | CLIP ConvNeXt-L | COCO-Stuff | 15.1 | 21.6 | 36.1 | 59.4 | 96.5 |
| MaskCLIP++ | CLIP ViT-L/14 | COCO-Stuff | **16.8** | **23.9** | **38.2** | 62.5 | 96.8 |

Panoptic (Lin et al., 2014) and evaluated on ADE20K (Zhou et al., 2017) using mIoU. As shown in Table 3, open-vocabulary mask generators do not significantly outperform closed-vocabulary mask generators in terms of segmentation performance ($P_s$). And using our fine-tuned CLIP for mask classification can significantly improve the segmentation performance ($P_c$) of all mask generators. This suggests that both open-vocabulary and closed-vocabulary mask generators possess an inherent category-independent mask generation capability that has yet to be fully harnessed. Furthermore, integrating the classification scores of the mask generator on training categories, as described in equation 4, can slightly enhance performance. For the hyperparameter $\gamma$, empirically, the worse the mask classification quality $P_s$ of the mask generator is, the smaller $\gamma$ can be set.

## 4.3 ENHANCEMENT OF MASK CLASSIFICATION BY MASKCLIP++

We apply our method to CLIP with various architectures to demonstrate its universality. The experimental settings are described in Section 4.1. Table 4 presents the types of CLIP architectures, the parts that were fine-tuned, and the corresponding number of fine-tuned parameters. Compared to the original CLIP, our method improved mIoU by 10.1%-16.8%, PQ by 5.4%-9.9% and AP by 1.6%-4.6% on ADE20K(Zhou et al., 2017) when using COCO Panoptic (Lin et al., 2014) for fine-tuning. Additionally, we demonstrate significant performance improvements of the original CLIP across different input resolutions, as shown in Figure 6 in Appendix C.

## 4.4 COMPARISON WITH PREVIOUS METHODS

**Open-vocabulary semantic segmentation.** There are notable differences in the training settings compared to Section 4.1 to be consistent with most previous open vocabulary segmentation works. We fine-tune CLIP on COCO-Stuff (Caesar et al., 2018), which shares the same images as COCO Panoptic (Lin et al., 2014) but defines 171 categories. We report mIoU performance on the validation sets of the following datasets: ADE-20K (Zhou et al., 2017) with 847 classes (A-847) and 150 classes (A-150), PASCAL-Context (Mottaghi et al., 2014) with 459 classes (PC-459) and 59 classes (PC-59), and PASCAL-VOC (Everingham et al., 2010) with 20 foreground classes (PAS-20).

Table 6: Comparisons with previous methods on Open-Vocabulary Panoptic and Instance Segmentation tasks. All methods are trained on COCO Panoptic (Lin et al., 2014) and evaluated on ADE20K (Zhou et al., 2017).

| Method | VLM | PQ | AP |
|---|---|---|---|
| FreeSeg (Qin et al., 2023) | CLIP ViT-B/16 | 16.3 | 6.5 |
| FC-CLIP (Yu et al., 2024) | CLIP ConvNeXt-B | 23.1 | 13.1 |
| MaskCLIP++ | CLIP ConvNeXt-B | **24.5** | **13.6** |
| MaskCLIP (Ding et al., 2023) | CLIP ViT-L/14 | 15.1 | 6.2 |
| MasQCLIP (Xu et al., 2023d) | CLIP ViT-L/14 | 23.3 | - |
| ODISE (Xu et al., 2023a) | CLIP ViT-L/14 | 22.6 | 14.4 |
| FC-CLIP (Yu et al., 2024) | CLIP ConvNeXt-L | 26.8 | 16.8 |
| MAFT+ (Jiao et al., 2024) | CLIP ConvNeXt-L | 27.1 | - |
| MaskCLIP++ | CLIP ViT-L/14 | **27.3** | **17.0** |

Table 7: Comparison of CLIP's mask classification ability in different methods. The same ViT-L/14 architecture of CLIP and the same mask generator are used. The first three rows are initialized by OpenAI-CLIP, and the last three rows are initialized by EVA-CLIP.

| Method | PC-459 | A-150 | PC-59 | Stuff |
|---|---|---|---|---|
| OpenAI | 9.3 | 17.6 | 27.4 | 18.0 |
| CAT-Seg | 19.6 | 34.1 | 57.3 | 45.6 |
| MaskCLIP++ | **21.3** | **34.8** | **59.3** | **47.2** |
| EVA02 | 11.1 | 20.1 | 26.0 | 18.3 |
| CLIPSelf | 13.0 | 22.9 | 39.1 | 27.1 |
| MaskCLIP++ | **24.2** | **37.8** | **62.3** | **50.6** |

Table 8: Performance improvements achieved when combined with several unsupervised open-vocabulary semantic segmentation methods that do not rely on a mask generator. OpenAI CLIP ViT-B/16 are used. "Citys" means cityscapes dataset (Cordts et al., 2016).

| Method | A-847 | PC-459 | A-150 | PC-59 | Stuff | Citys |
|---|---|---|---|---|---|---|
| SCLIP (Wang et al., 2024a) | - | 7.7 | 16.1 | 34.2 | 22.4 | 32.2 |
| SCLIP w/ MaskCLIP++ | - | 9.5 | 19.3 | 40.8 | 29.0 | 35.9 |
| ClearCLIP (Lan et al., 2024a) | - | 7.9 | 16.7 | 36.0 | 23.9 | 30.0 |
| ClearCLIP w/ MaskCLIP++ | - | 9.6 | 19.2 | 40.8 | 30.4 | 33.5 |
| ProxyCLIP (Lan et al., 2024b) | 7.4 | 8.4 | 20.2 | 39.1 | 26.5 | 38.1 |
| ProxyCLIP w/ MaskCLIP++ | 8.0 | 9.9 | 23.2 | 44.6 | 34.2 | 42.4 |

For comparisons with previous works using base-level VLM models, we fine-tune CLIP ConvNeXt-B with previous settings. For comparisons with works using larger-level VLM models, we fine-tune CLIP ViT-L/14. Apart from the previous settings, we also fine-tune the input projections of the attention in the CLIP text encoder following CAT-Seg (Cho et al., 2024). We use the same mask generator as MAFT+ (Jiao et al., 2024) during inference and set $\gamma = 0.4$ for CLIP ConvNeXt-B and $\gamma = 0.1$ for CLIP ViT-L/14.

Table 5 shows that our base model achieves the best performance on A847, A150 and PC-59, and our large model achieves the best performance on A-847, PC-459, and A-150.

**Open-vocabulary panoptic and instance segmentation.** Following the settings of previous works, we fine-tune CLIP on COCO Panoptic (Lin et al., 2014). All methods are evaluated on ADE20K, with PQ measuring panoptic segmentation performance and AP measuring instance segmentation performance. We use the same mask generator as FC-CLIP (Yu et al., 2024) and do not use the ensemble strategy ($\gamma = 0.0$). Table 6 shows that our method achieves the best performance.

**Inference with the same masks.** Considering that some competitive methods in Table 5 do not explicitly use a mask generator and fine-tune CLIP in alternative ways, we test the mask classification performance of CLIPs from various sources under the same generated masks. Since the PSM module we trained may not be compatible with their CLIP, we remove the PSM during inference. The ensemble strategy with the mask generator is also excluded. As shown in Table 7, our fine-tuned CLIPs outperform ones from other methods in mask classification.

**Inferece without mask generators.** Because the goal of our fine-tuning is to enhance the mask classification ability of CLIP, which indirectly enhances the local representation between different patch representations of the CLIP. As shown in Table 8, after applying to some methods that do not require training and additional mask generators, further performance improvements can be obtained.

**Trade-off between training cost and performance.** Figure 4 shows the trade-off between training cost and performance of different open-vocabulary segmentation methods. As our method does not use the mask generator or teacher model during training but only a small number of GT masks

Table 9: The impact of the amount of data used for fine-tuning. "data size" indicates the proportion of data used for fine-tuning relative to the original COCO-Stuff training set, and the fine-tuned model is CLIP-ConvNeXt-B.

| Data size | A-847 | PC-459 | A-150 | PC-59 | Stuff |
|-----------|-------|--------|-------|-------|-------|
| 100% | 14.5 | 18.7 | 35.4 | 59.1 | 47.1 |
| 10% | 14.2 | 19.0 | 34.9 | 59.2 | 47.2 |
| 5% | 13.9 | 18.7 | 35.2 | 59.6 | 47.3 |
| 1% | 13.8 | 19.1 | 35.1 | 58.5 | 47.0 |
| 0.1% | 12.8 | 18.6 | 34.1 | 58.2 | 46.1 |

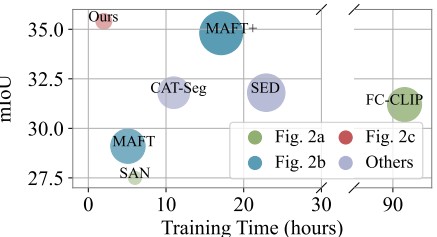

Figure 4: Trade-off between training cost and performance of different OVS methods. Radius size represents the memory usage during training. Using CLIP with base size for comparison.

instead of a large number of generated masks and has only one classification loss, it converges quickly, has lower memory usage, and achieves better performance.

**Data efficiency.** Since the goal of our fine-tuning is to transfer CLIP's open-vocabulary recognition capabilities to mask regions, rather than learning category-specific pixel groupings from segmentation datasets, our method can effectively fine-tune using a smaller amount of image segmentation data. We randomly sampled varying proportions of data from the COCO-Stuff training set for fine-tuning, using the same configuration as the base model in Table 5. As shown in Table 9, even with only 0.1% of the data (approximately 118 images with annotations), the fine-tuned model still demonstrates competitive performance.

**Qualitative comparison of segmentation results.** Figure 7 shows the qualitative results on the Open-Vocabulary Semantic Segmentation task. Using the mask generator from MAFT+ (Jiao et al., 2024), we compare the segmentation results of the original CLIP, MAFT+, and our method. Compared to the original CLIP, our method achieves more accurate recognition of local images. Compared to MAFT+, our method retains more of the original CLIP's knowledge without generating too many erroneous biases. Our method also predicts relatively correct semantics for some objects that are not annotated in the GT.

## 5 CONCLUSIONS

In this paper, we point that using generated masks hinders CLIP's regional image-text alignment in previous OVS work. And there is limited room for improving mask generation compared to mask classification. We also reveal that attempts to update CLIP's embedding vectors disrupt its original pre-alignment properties, resulting in new alignments that function only on the training data. Based on these analysis, we propose the MaskCLIP++, which utilizes ground truth masks as prior to fine-tune CLIP, and models similarity under a consistency alignment constraint to mitigate overfitting. Our method significantly enhances CLIP's mask recognition capabilities across various architectures. By integrating mask generators from prior work, we achieve comparable or superior performance on semantic segmentation, panoptic segmentation, and instance segmentation tasks with lower training costs. Overall, we adapt CLIP more effectively to open-vocabulary segmentation tasks through the use of image segmentation data and lightweight fine-tuning.

## REPRODUCIBILITY STATEMENT

In Section 4.2, we provide details on the source of pre-trained model, training data, and training strategies. Additional details regarding experiments on open-vocabulary semantic segmentation are given in Section 4.4. Appendix B provides detailed descriptions of the datasets used for training and evaluation. Appendix C provides additional experiments, including detailed settings of the model. Our model and code will be made publicly available.

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

# APPENDIX

## A  LIMITATIONS AND FUTURE WORK

First, the image masks must be downsampled to match the size of CLIP's final feature map, which undergoes a downsampling factor of 14 to 32 times. Consequently, some generated small object masks may not be recognizable by MaskCLIP++.

Second, the mask generators employed during inference incur negligible computational overhead. As shown in Table 3, heavier backbones seem to enhance classification capabilities of their own ($P_s$), but do not significantly improve open-vocabulary segmentation performance when combined with our fine-tuned CLIP ($P_c$). Therefore, lightweight, category-agnostic mask generators continue to represent a promising research direction.

Last, our reliance on GT masks somewhat limits our ability to scale up at the data level. Finding ways to utilize more image-text pairs or image-mask pairs for fine-tuning remains an open question.

## B  DATASETS OF TRAINING AND EVALUATION

The image segmentation datasets we used to train are COCO Panoptic (Lin et al., 2014) or COCO-Stuff (Caesar et al., 2018). For evaluation, we utilize the ADE20K (Zhou et al., 2017), PASCAL VOC (Everingham et al., 2010), and PASCAL Context (Mottaghi et al., 2014) datasets. A brief introduction to these datasets is provided below.

**COCO Panoptic.** It contains 164K images with 133 annotated classes, comprising 80 "thing" categories and 53 "stuff" categories. The "thing" categories are annotated at the instance level, with different instances of the same category using separate masks within a single image. In contrast, the "stuff" categories are semantically annotated, where all regions belonging to the same category share a single mask in an image.

**COCO Stuff.** It shares the same training images with COCO Panoptic. These images are annotated for semantic segmentation into 171 categories.

**ADE20K.** It contains 2,000 images for validation with two types of annotations: one for panoptic or semantic segmentation with 150 categories (dubbed as A-150), and the other for semantic segmentation with 847 categories (dubbed as A-847). Among the 150 categories defined in ADE20K, 64 appear in COCO Panoptic, while 86 do not. The names of these categories are detailed in Table 10.

**PASCAL VOC.** It includes 1,449 images for testing with 20 annotated foreground classes for semantic segmentation.

**PASCAL Context.** It is a dataset for semantic understanding that contains 5,000 validation images. Two versions are employed for semantic segmentation: one with 59 frequently used classes (referred to as PC-59) and another with the complete set of 459 classes (referred to as PC-459).

## C  ADDITIONAL EXPERIMENTS

### C.1  ADDITIONAL ILLUSTRATIONS

**Implementation details of Figure 1** We use mask generation and mask classification as the inference modes, with Figure 1(a) and Figure 1(b) presenting the performance upper bounds of the ideal mask generator and ideal mask classifier, respectively. The ideal mask generator is implemented by

Table 10: Categories in ADE20K that are present in COCO Panoptic (seen) or are absent from COCO Panoptic (unseen).

| Split | Name |
|---|---|
| Seen (64) | *wall, building, sky, floor, tree, ceiling, road, bed, windowpane, grass, cabinet, sidewalk, person, door, table, mountain, plant, curtain, chair, car, water, sofa, shelf, sea, mirror, rug, fence, rock, lamp, counter, sand, sink, refrigerator, stairs, pillow, river, bridge, toilet, flower, book, bench, palm tree, boat, bus, towel, light bulb, truck, television receiver, airplane, apparel, bottle, tent, oven, food, microwave, plant pots, animal, bicycle, blanket, vase, traffic light, plate, cup, clock* |
| Unseen (86) | *earth, painting, house exterior, field, armchair, seat, desk, wardrobe, bathtub, railing, cushion, pedestal, box, column, signboard, chest of drawers, skyscraper, fireplace, grandstand, path, runway, case, pool table, screen door, stairway, bookcase, window screen, coffee table, hill, countertop, stove, kitchen island, computer, swivel chair, arcade machine, hovel, tower, chandelier, awning, streetlight, booth, dirt track, pole, land, bannister, escalator, ottoman, buffet, poster, stage, van, ship, fountain, conveyer belt, washer, plaything, swimming pool, stool, barrel, basket, waterfall, bag, minibike, cradle, step, tank, trade name, lake, dishwasher, projection screen, sculpture, exhaust hood, sconce, tray, ashcan, ceiling fan, pier, crt screen, monitor, bulletin board, shower, radiator, canopy, flag, bar, ball* |

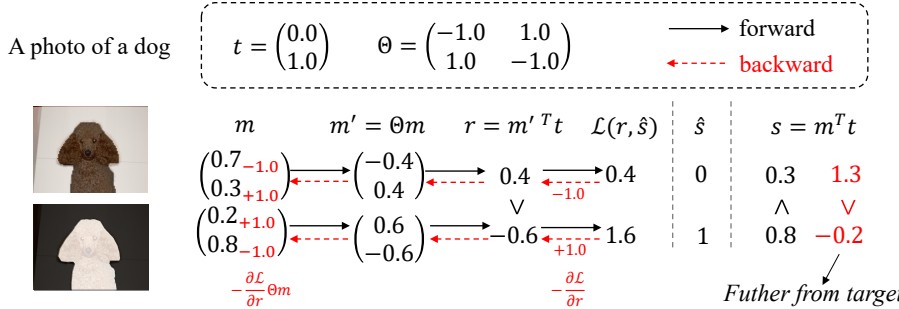

Figure 5: A toy example: Because the inconsistency alignment between $s$ (similarity before $\Theta$) and $r$ (similarity after $\Theta$), after a round of gradient descent, $s$ is further from the target $\hat{s}$.

directly using the ground truth masks. To implement the ideal mask classifier, we compute the maximum matching between the generated masks and ground truth masks based on IoU, then assign the corresponding GT categories to the matched masks and combine them into the output segmentation maps.

**Phenomenon of inconsistency alignment.** We first use a toy example to illustrate the potential consequences of destroying alignment consistency. Assume that we use the parameter $\Theta$ to update the mask embeddings, changing the original similarity from $s$ to $r$, and then optimize towards the target. However, due to the lack of explicit constraints on $\Theta$, there may be a situation as shown in Figure 5, where the order of $s$ and $r$ is inconsistent, resulting in correct optimization on $r$ but incorrect optimization on $s$. For simplicity, $\mathcal{L}$ in Figure 5 denotes an L1 loss without softmax, and only mask embeddings $m$ are updated in one round of gradient descent.

Futuremore, we use Table 11 as a supplement to Table 1 to illustrate the extent to which different PSM structures destroy the consistency alignment constraint. For models fine-tuned with various PSMs, we test by retaining or removing the PSM module, and compare the performance differences to assess whether the alignment consistency is maintained before and after the PSM intervention. The results show that only the three blue PSM structures largely satisfy the consistency alignment constraint, while the three red PSM structures severely disrupt it.

Table 11: The phenomenon of consistency alignment.

| PSM | test | mIoU(S) | mIoU(U) | mIoU |
|---|---|---|---|---|
| - | - | 32.7 | 16.3 | 23.3 |
| $\langle \text{Linear}(E_m), E_t \rangle$ | ✗ | 37.7 | 20.2 | 27.7 |
| | ✓ | 32.4 | 1.9 | 14.9 |
| $\langle E_m, \text{Linear}(E_t) \rangle$ | ✗ | 37.6 | 20.3 | 27.6 |
| | ✓ | 32.5 | 1.8 | 14.9 |
| $\text{Linear}\langle E_m, E_t \rangle$ | ✗ | 45.2 | 24.6 | 33.4 |
| | ✓ | 45.4 | 25.7 | 34.1 |
| $\langle E_m, \text{Norm}(P_t) \rangle$ | ✗ | 39.5 | 21.4 | 29.1 |
| | ✓ | 31.9 | 4.5 | 16.2 |
| $\langle E_m, \text{Norm}(E_t) \rangle$ | ✗ | 45.8 | 24.4 | 33.5 |
| | ✓ | 46.2 | 25.2 | 34.1 |
| $\langle E_m, \text{Norm}(E_t + P_t) \rangle$ | ✗ | 44.0 | 24.1 | 32.6 |
| | ✓ | 46.3 | 26.7 | 35.1 |

Table 12: Performance of MaskCLIP++ on closed-vocabulary image segmentation tasks.

| CLIP arch. | Dataset | mIoU | PQ | AP |
|---|---|---|---|---|
| ConvNeXt-B | COCO-Stuff | 47.1 | - | - |
| ViT-L/14 | COCO-Stuff | 51.0 | - | - |
| ConvNeXt-B | COCO-Panoptic | 57.6 | 46.1 | 34.4 |
| ViT-L/14 | COCO-Panoptic | 63.3 | 52.3 | 42.3 |

**The performance of closed-vocabulary image segmentation.** We present the performance of MaskCLIP++ on closed-vocabulary image segmentation in Table 12, with the same configuration as in Table 5 and Table 6, respectively.

## C.2 ADDITIONAL ABLATIONS

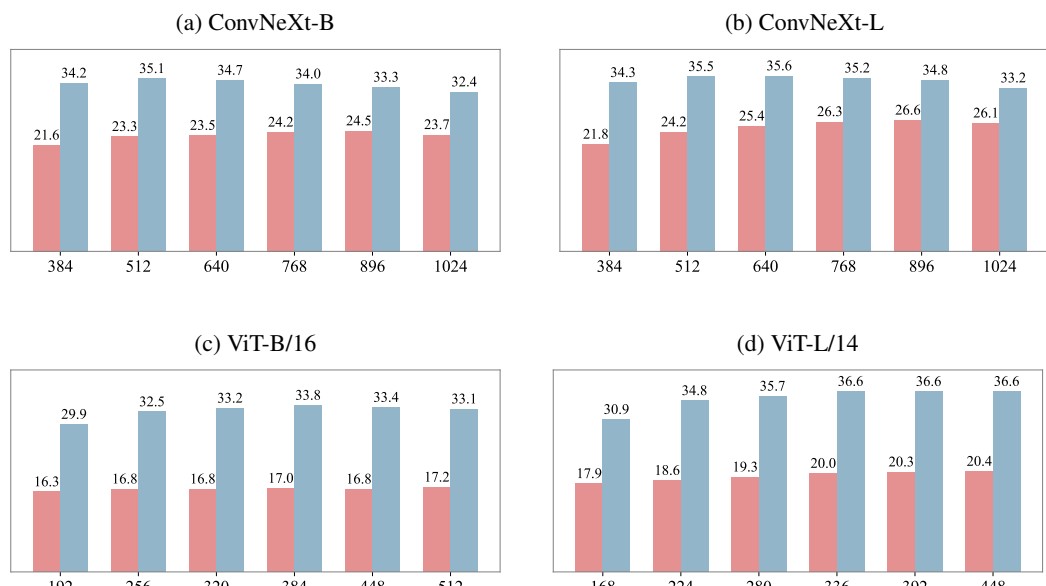

Figure 6: Under various input sizes of different CLIP models, our method (blue bars) has significant improvements over the original CLIP (red bars). The input size of the mask generator remains unchanged. All models are fine-tuned on COCO Panoptic and evaluated on ADE20K with mIoU reported.

**Impact of CLIP input size.** Considering that input image size often has a significant impact on the mIoU of semantic segmentation models, we compared the performance of our method with the original CLIP under different input sizes. As shown in Figure 6, with a fixed input size for the mask generator, our fine-tuned CLIP consistently outperforms the original CLIP across various input sizes for open-vocabulary semantic segmentation.

Table 13: Impact of integrating image context information into mask embeddings $E_m$. The implementation varies based on the original CLIP's pooling method.

(a) The case of CLIP using global average pooling, taking ConvNeXt-B as an example. $\Psi$ is a placeholder for image context information, and $P_t$ defined in equation 3 can be rewritten as $P_t = \text{Attn}(q(E_t), k(\Psi), v(\Psi))$. $P_t$ interacts with $E_m$ in the PSM stage to increase the utilization of image context information.

| Method in PSM | Image Context | mIoU |
|---|---|---|
| No $\Phi$ | ✗ | 34.1 |
| $\Phi = E_i$ | ✓ | 34.5 |
| $\Phi = E_p$ | ✓ | **35.1** |

(b) The case of CLIP using attention pooling, taking ViT-B/16 as an example. $\alpha$ is defined in equation 2. The bigger $\alpha$ is, the less the semantic similarity of the image context is utilized. The learnable $\alpha$ is initialized from $e^{-5}$ and learned in the logarithmic space.

| Method in CLIP | Image Context | mIoU |
|---|---|---|
| $\alpha = e^5$ | ✗ | 31.1 |
| $\alpha = e^{-5}$ | ✓ | 33.5 |
| learnable $\alpha$ | ✓ | **33.8** |

Table 14: Choice of Loss. "Re." means the loss is reweighted by categories. CLIP ConvNeXt-B is used.

| Prior | Loss | Re. | mIoU(S) | mIoU(U) | mIoU |
|---|---|---|---|---|---|
| Pixel | CE | ✗ | 39.5 | 21.2 | 29.0 |
| | CE | ✓ | 44.3 | 23.4 | 32.8 |
| Instance Mask | BCE | ✗ | 45.7 | 24.5 | 33.5 |
| | BCE | ✓ | 46.3 | 25.3 | 34.3 |
| | CE | ✗ | 45.5 | 26.0 | 34.4 |
| | CE | ✓ | **46.3** | **26.7** | **35.1** |

Table 15: The impact of the position of mask prior insertion for ViT-B/16 CLIP. "Prior pos." denotes the insert position relative to the total number of attention blocks.

| Prior pos. | mIoU | PQ |
|---|---|---|
| -1 | 33.3 | 24.1 |
| -2 | 33.5 | **24.5** |
| -3 | **33.8** | 24.4 |
| -4 | 33.5 | 24.4 |

Overall, segmentation performance initially increases and then decreases as input size grows. Since our fine-tuned CLIP can better recognize local image features, the optimal input size tends to be smaller than that of the original CLIP. Our framework allows us to maintain the input size for the mask generator while reducing the input size for mask recognition, thereby enhancing performance and reducing computational costs.

**Impact of integrating image context information into mask representation.** Averaging CLIP features over perfectly precise mask regions is not conducive to extracting mask representations with stronger recognition capabilities, as recognition tasks typically require contextual information and focused attention on target internal regions. Here, we distinguish how we address this common issue using CLIP's original pooling method. For CLIP models utilizing Attention Pooling (e.g., ViT-B/16), the hyperparameter $\alpha$ controls the degree of contextual information incorporated in the mask representation. As shown in Table 13b, smaller $\alpha$ values, indicating more extensive use of contextual information, lead to better performance. For CLIP models employing global average pooling (e.g. ConvNeXt-B), as shown in Table 13a, using global image embeddings $E_i$ or patch embeddings $E_p$ to construct pseudo-text embeddings $P_t$ in the PSM stage enhances the utilization of image context information to some extent, thereby improving performance.

**Choice of Loss.** Table 14 presents the performance differences observed when fine-tuning CLIP ConvNeXt-B with various loss functions. Training with COCO Panoptic (Lin et al., 2014) and evaluating on ADE20K (Zhou et al., 2017). When the training labels for a single image include multiple instances of the same category, such as pixels or instance masks, we re-weight the loss by categories to prevent bias toward more common classes during optimization. We adopt a class-reweighted cross-entropy loss for all our experiments in the main text.

**Position of the mask prior insertion.** Due to the repetitive nature of the ViT architecture, the position of mask prior insertion in CLIP models based on such architectures is not absolute. Following the approach of Xu et al. (2023c) for constructing attention masks, we can insert the mask prior at an earlier position than the last attention block. Table 15 takes ViT-B/16 as an example, demonstrating that inserting the mask prior at the third-to-last layer yields better performance. Therefore, in Table 4, the mask prior is inserted at the third-to-last layer for all ViT-based CLIP models.

**Qualitative results.** Figure 7 compares the visual results of open-vocabulary semantic segmentation using different methods with the same mask generator. We show more qualitative results of semantic- and instance level of open-vocabulary segmentation in Figure 8.

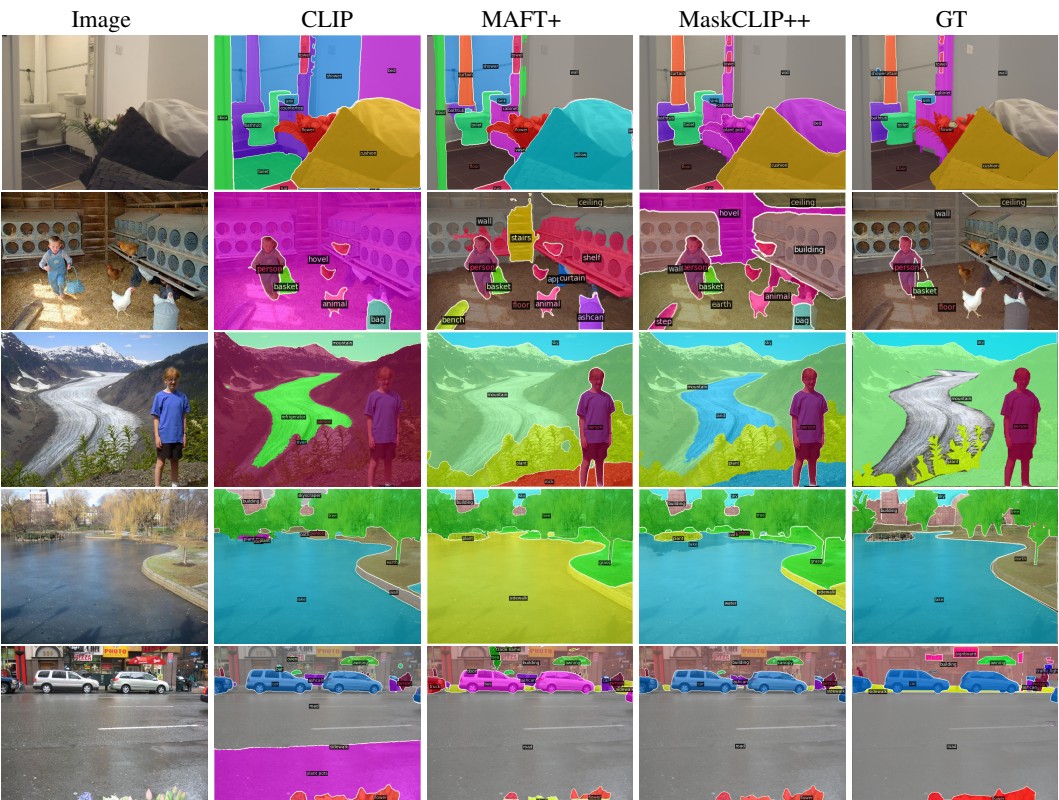

Figure 7: Visualizations of open-vocabulary semantic segmentation on ADE20K (Zhou et al., 2017). From left to right are the input image, the result using the original CLIP, the result of MAFT+ (Jiao et al., 2024), the result using MaskCLIP++, and the ground truth. The same mask generator is used for CLIP, MAFT+ and MaskCLIP++.

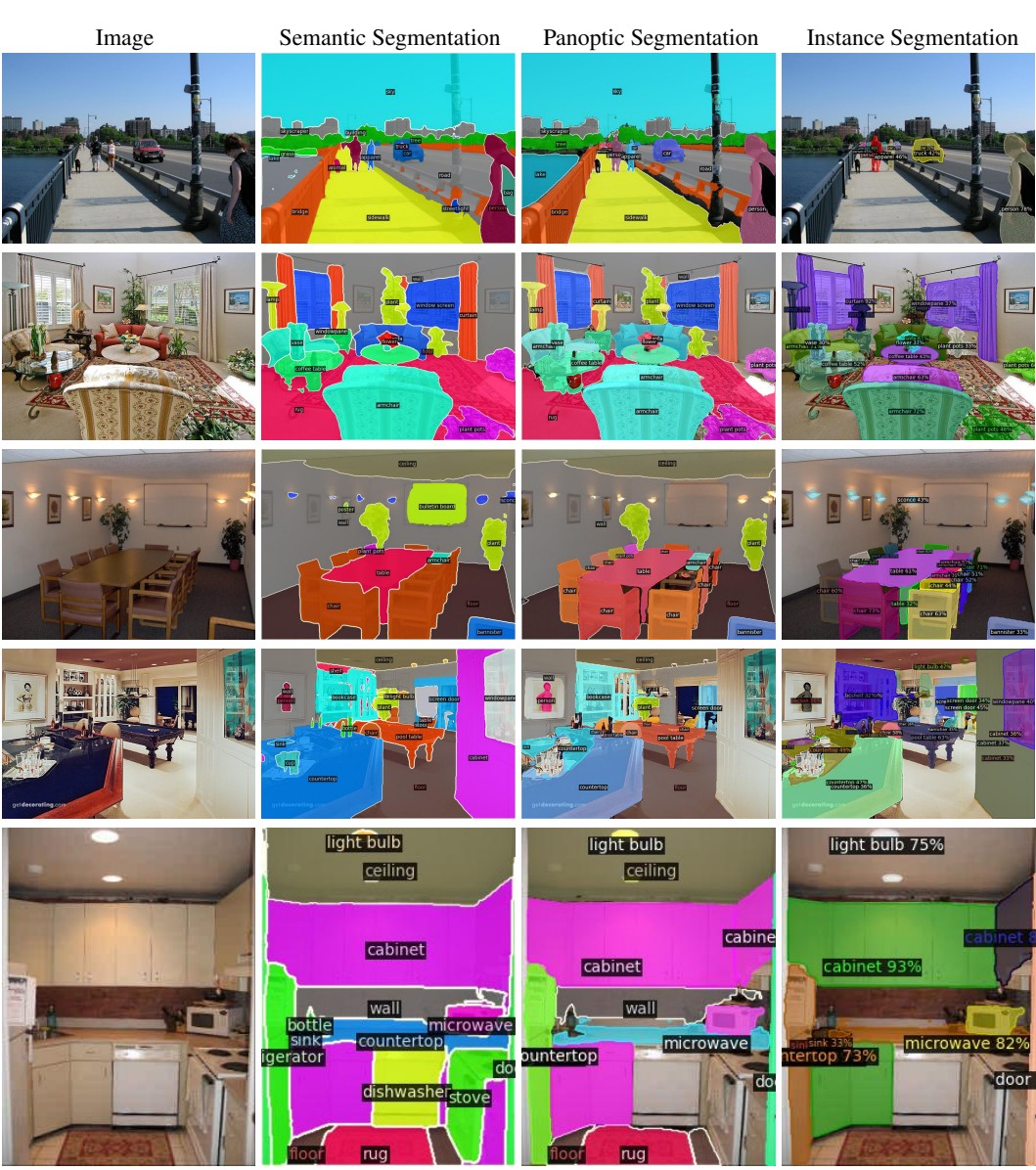

Figure 8: More visualizations of semantic- and instance-level of open-vocabulary segmentation. We use mask generator from FC-CLIP (Yu et al., 2024) with ConvNeXt-L backbone. And use CLIP ViT-L/14 fine-tuned on COCO Panoptic (Lin et al., 2014). Images are from ADE20K (Zhou et al., 2017). The confidence threshold for instance segmentation is 0.3.