# OpenReview forum: "MaskCLIP++: A Mask-Based CLIP Fine-tuning Framework for Open-Vocabulary Image Segmentation"
_ICLR.cc/2025/Conference — Submitted to ICLR 2025_

### Official Review · Reviewer_LpvX · 2024-10-27

**Soundness:** 4
**Presentation:** 3
**Contribution:** 3
**Rating:** 8
**Confidence:** 5

**Summary:**

This paper proposes MaskCLIP++ which uses ground-truth masks during training and can adapt many off-the-shelf segmentation networks during inference. The proposed method significantly reduces the training time while achieving SOTA performances on standard OVSS and OVPS benchmarks.

**Strengths:**

1. The proposed method, MaskCLIP++, reaches SOTA performances on standard open-vocabulary benchmarks while requiring significantly less training cost and can adapt many off-the-shelf segmentation network (e.g. Mask2Former and FC-CLIP) during inference.
2. The ablation studies presented in this article are thorough and well-structured, offering valuable insights into the impact of each design choice.
3. The proposed module about consistency alignment which enables training with ground-truth masks and inferencing with off-the-shelf segmentor seems novel and interesting to me.

**Weaknesses:**

1. The experiment results, though SOTA, is only marginally higher than previous SOTA e.g. CAT-Seg and MAFT+. The improvement seems marginal when the number of categories is smaller (e.g. ADE20K150, PC59, PC20). Moreover, I failed to find the performances on closed-vocabulary setting i.e. mIoU/PQ on COCO-stuff/COCO-Panoptic, which the authors should consider reporting these results in supplementary materials.
2. Is the proposed method capable of adapting any class-agnostic mask generator for inference? a) a simple experiments may be to use ground-truth but class-agnostic masks which can serve an upper bound for class-agnostic mask generator; b) if yes, is it possible to extend this work by combining SAM?
3. What is the inference efficiency in terms of FLOPs in comparison to previous methods?
4. CAT-Seg also benchmarks its approach on the MESS datasets [1], which include domain-specific data distinct from the training source. To further demonstrate the strengths of the proposed method, the authors might consider conducting experiments and providing comparative results with CAT-Seg on this benchmark.

[1] Benedikt et al. What a mess: Multi-domain evaluation of zero-shot semantic segmentation. NeurIPS 2023

**Questions:**

1. I would like to learn more about the detailed implementation of class inclusion/exclusion in Equation(4). For example, COCO makes a fine-grained distinction among wall-stone, wall-brick, etc but ADE20K considers the super-category wall. From Table 7 in Appendix, it seems that wall is considered as seen category. Is this classification manual or based on text overlapping of super-categories? What are potential impacts that super-categories appear in training source but fine-grained classification is needed in zero-shot datasets? Some qualitative comparisons would be helpful for understanding.
2. I am curious about the method's name "MaskCLIP++". Is there a relationship to MaskCLIP, which employs a pre-trained class-agnostic mask proposal network and uses predicted masks to refine attention masks?

The following are some typos that do not affect my ratings:
1. The paper *Learning to generate text-grounded mask for open-world semantic segmentation from only image-text pairs* appears twice in References.

---

> ### Author Response · Authors · 2024-11-22
>
> We sincerely thank reviewer LpvX for acknowledging our presentation and contributions, as well as for pointing out typos in our manuscript. We will provide further clarification regarding the concerns raised by Reviewer LpvX.
>
>
> (1) **Closed-Vocabulary Performance:** The table below summarizes the corresponding performance of MaskCLIP++ on COCO-Stuff and COCO-Panoptic, as reported in Tables 5 and 6.
>
> |VLM|train dataset|mIoU | PQ | AP |
> |:---:|:---:|:---:|:---:|:---:|
> |CLIP-ConvNeXt-B|COCO-Stuff|47.1 | -  | -  |
> |CLIP-ViT-L/14  |COCO-Stuff|51.0 | -  | -  |
> |CLIP-ConvNeXt-B|COCO-Panoptic|57.6 |46.1 |34.4 |
> |CLIP-ViT-L/14  |COCO-Panoptic|63.3 |52.3 |42.3 |
>
>
> (2) **Regarding Additional Mask Generators:** Because the mask generator and CLIP operate as two independent modules in our framework, any mask generator capable of producing a set of mask proposals from an input image can be integrated into the inference process. This adaptability is demonstrated in both Fig. 1b and Table 3 of the main text. As a complement to Table 3, we further evaluate the performance upper bound using ground truth (GT) masks. Additionally, we incorporate SAM (Segment Anything Model) in a simple way: Using SAM under its default configuration to provide mask proposals, then applying our model for mask classification. The results are presented in the table below, under the same experimental setup as Table 3:
>
> |Mask Generator| mIoU (from $P_c$) |
> |:------------:|:-----------------:|
> | GT masks     | 44.2              |
> | SAM-B        | 21.8              |
> | SAM-H        | 31.0              |
>
> Although SAM generates accurate masks, it is observed that under the default configuration, SAM has a lower mask recall compared to Mask2Former, leading to inferior overall performance.
>
>
>
> (3) **Regarding FLOPs**: Existing FLOPs calculation tools are primarily designed for end-to-end network computations. However, in various OVS inference modes, **certain custom computations within the models are not easily accounted for automatically**. Therefore, we provide below the average inference latency on ADE20K for recent methods tested on the same device (a single RTX 3090, using the best-performing configurations from Table 5), as shown in the table below:
>
> |Method|A-150|Latency(ms)|
> |:---:|:---:|:---:|
> |FC-CLIP|34.1 |26.7  |
> |CAT-Seg|37.9 |47.2  |
> |Ours   |38.2 |30.0  |
>
>
> Due to the complete decoupling of CLIP and the mask generator, our inference typically incurs additional computation in CLIP-V compared to methods where the mask generator and CLIP share a backbone (e.g. FC-CLIP).
>
> However, it is important to note that the shared-backbone design may limits the applicability across various VLMs: For example,
> FC-CLIP does not perform well with ViT-based CLIP architectures. In contrast, the decoupled-backbone inference framework supports both CNN and ViT architectures of CLIP, and offering greater flexibility by allowing the mask generator to be plug-and-play.
>
> (4) **On the Implementation of Class Inclusion/Exclusion**:
> Integrating CLIP and mask generator classification scores based on vocabulary overlap with the training set is a common technique in prior mask-based OVS work [1,2,3]. In our approach, we inherit the implementation of this strategy from FC-CLIP but apply it only to seen categories. Below are our responses to related questions:
>
> - **Is this classification manual or based on text overlapping of super-categories?**
>   It is manual. FC-CLIP expands vocabulary with synonyms for several datasets. For example, in COCO-Stuff, "wall-other" is expanded to include "wall," which consequently marks "wall" in ADE as seen and uses the score of "wall-other" from the mask generator during ensemble.
>
> - **What are the potential impacts of super-categories appearing in the training source while fine-grained classification is required in zero-shot datasets?**
>   Since overlaps are manually specified, the impact depends on whether the test category appears in the synonym list of any training category.
>
> - **What is the extent of the impact?**
>   Our use of the ensemble aims to leverage knowledge from the mask generator appropriately, as the additional computational cost is negligible. **However, this is not the focus of our method, and its minimal impact on benchmarks in Table 5.** When the relationship between test and training categories is unclear, we recommend disabling this strategy.
>
>
> |Ensemble on seen classses| A-847 | PC-459 | A-150 | PC-59 | PC-20 |
> |:------:|:------:|:------:|:------:|:------:|:------:|
> |   Yes  |  16.8 |  23.9  | 38.2  | 62.5  | 96.8 |
> |   No   |  16.8 |  24.2  | 37.8  | 62.3  | 96.8 |

---

> > ### Author Response · Authors · 2024-11-22
> >
> > (5) **On the Evaluation on MESS Data**:  The benchmarks used in our paper are the standard metrics commonly adopted by existing OVS methods. Since CAT-Seg did not disclose the details of their evaluation on MESS in their open-source code and a significant portion of the 22 datasets in MESS are difficult to access, we do not plan to complete the evaluation on MESS during the discussion phase.
> >
> > (6) **Relationship with MaskCLIP**:  Our method's name may cause some misunderstanding, as clarified in our response to Reviewer mMjr: *“Clarification regarding the relationship with MaskCLIP.”* To put it simply, our approach is not a next version of any previously named "MaskCLIP" method.
> >
> >
> > We sincerely thank Reviewer LpvX for the valuable feedback and thoughtful suggestions. We look forward to engaging in further discussions. In the revised paper, all changes will be highlighted in blue and carefully aligned with our responses.
> >
> > ---
> >
> > [1] Xu, Mengde, et al. "A simple baseline for open-vocabulary semantic segmentation with pre-trained vision-language model." ECCV, 2022.
> >
> > [2] Xu, Jiarui, et al. "Open-vocabulary panoptic segmentation with text-to-image diffusion models." CVPR. 2023.
> >
> > [3] Yu, Qihang, et al. "Convolutions die hard: Open-vocabulary segmentation with single frozen convolutional clip." NeurlPS, 2023.

---

> > > ### Comment · Reviewer_LpvX · 2024-11-24
> > >
> > > Thanks authors for their efforts. After rebuttal, the experiments and discussions have addressed my concerns. Therefore, I increased my rating to 8.

---

> > > > ### Author Response · Authors · 2024-11-24
> > > >
> > > > We sincerely thank you for recognition of our work and the responses we provided.

---

### Official Review · Reviewer_n1UQ · 2024-10-29

**Soundness:** 3
**Presentation:** 2
**Contribution:** 2
**Rating:** 5
**Confidence:** 4

**Summary:**

This paper presents MaskCLIP++, a masked-based CLIP fine-tuning framework designed for open-vocabulary semantic segmentation. It identifies a performance bottleneck in mask classification accuracy and proposes a consistency alignment module to improve local image-text alignment while mitigating overfitting issues. Experimental results on several benchmarks demonstrate that MaskCLIP++ outperforms existing methods in open-vocabulary semantic, panoptic and instance segmentation  tasks. Additionally, extensive ablation studies are conducted to validate the effectiveness of the proposed framework.

**Strengths:**

- **Motivation:** The motivation for this work is clearly articulated in the introduction, providing a solid foundation for the research.
- **Quality:** The techniques employed in this paper are sound and well-presented, although some visual illustrations would enhance clarity and understanding.
- **Experiment:** This work conducts extensive experiments to thoroughly validate all aspects of the proposed methods, demonstrating a robust evaluation process.

**Weaknesses:**

- **Originality:** The primary contribution of this work is the proposed PSM, which enhances local image-text alignment while mitigating overfitting issues. However, this consistency alignment is reminiscent of the cost volume aggregation used in CAT-Seg, which considerably diminishes the novelty of the approach.

- **Presentation:** The presentation of this work requires improvement. For instance, Figure 3 illustrates the detailed framework of MaskCLIP++, but the clarity of this illustration is insufficient. There is no detailed depiction of the Neck module or the PSM module. Furthermore, while the PSM contains several learnable components, their representations are not adequately explained. Additional unclear aspects are provided in the Questions section.

- **significance:** The performance of MaskCLIP++ is not sufficiently impressive. As indicated in Table 5, it achieves competitive results in most cases. For example, when compared with CAT-Seg, which utilizes a large VLM, MaskCLIP++ does not demonstrate superior performance despite the introduction of a mask generator and various hyperparameters.

**Questions:**

- The authors assert that this framework eliminates the need for a specific mask generator. However, this claim is unsubstantiated, as MaskCLIP++ still requires a specific mask generator during inference, and the choice of mask generator can significantly influence the final performance.
- The claim that the consistency alignment module can substantially reduce overfitting issues is not well-supported by the experimental results. There is no evidence presented regarding overfitting, making it questionable to attribute overall performance improvements to overcoming these issues..
- While the motivation is effectively articulated in the introduction, additional experimental results are needed to demonstrate how well this work enhances mask classification compared to the original mask generator. Since this work aggregates both the class probabilities of the fine-tuned CLIP and the mask generator, direct comparison becomes challenging.
- In lines 207-210 on page 4, the authors claim that once &r& is optimized to align with the target, the original similarity relationships become disrupted, necessitating the consistency alignment. However, this observation may instead indicate that the original similarity relationships are noisy and require improvement. This perspective aligns with findings in CAT-Seg, where the original similarity was also deemed noisy, leading to the use of aggregation on the similarity matrix.

---

> ### Author Response · Authors · 2024-11-22
>
> We appreciate Reviewer n1UQ’s recognition of the effectiveness of our work. We will address the concerns raised by n1UQ in further detail.
>
> (1) **Clarification on the Neck structure**: **Our goal is to improve the mask classification performance of CLIP across various architectures, which is why we do not illustrate a specific CLIP architecture in Fig. 3.** However, we have clarified in the first paragraph of Section 3.2 that the neck is part of CLIP-V and summarized its role in spatial dimension aggregation in Equation (1). We then provide a detailed classification discussion based on the original CLIP's pooling methods in the rest of the paragraph.
>
> (2) **Clarification on the PSM structure and consistency alignment**: We have provided the design principles and structural details of PSM in the second paragraph of Section 3.2. In summary, our PSM structure is very simple, often requiring just a single linear layer. For CLIP models originally using average pooling, we supplement global information during the PSM process as described in Eq. 3. **The key design principle behind these simple PSM structures—consistency alignment—is what we aim to emphasize in Fig. 3.**
>
> (3) **Clarification on "not needing a mask generator":** This paper operates within the inference paradigm of mask-based OVS. By analyzing the potential of mask generators and CLIP, we conclude that the bottleneck in CLIP's mask classification limits the full utilization of pretrained mask generators, and joint training may even negatively affect CLIP's mask classification performance. **Our innovation lies in the fine-tuning approach, without altering the inference paradigm of mask-based OVS.**
>
> (4) **Response to "the consistency alignment module can substantially reduce overfitting issues is not well-supported by the experimental results"**. In Table 1 and the first paragraph of Section 4.2, we demonstrate the importance of maintaining consistency alignment in PSM design for alleviating overfitting during fine-tuning. Furthermore, in our response to Reviewer gVpU under "Regarding the design principles of PSM," we expanded Table 1 with experiments that highlight the phenomena caused by severely violating consistency alignment.
>
> (5) **Response to "There is no evidence presented regarding overfitting"**: As described in the experimental setup in Section 4.1 and the first paragraph of Section 4.2, we partitioned the validation set categories into seen and unseen based on their overlap with the vocabulary of the training set. The baseline in Table 1 represents the mask classification performance of the original CLIP on seen and unseen categories. Evidence of overfitting is demonstrated by the significant drop in performance on unseen categories after fine-tuning, which falls far below the baseline.
>
> (6) **Response to "how well this work enhances mask classification compared to the original mask generator."**: In Table 3, the $P_s$ column reports the original performance of various mask generators, while the $P_c$ column presents the mask classification performance achieved using MaskCLIP++ on these mask generators. The subsequent $P_\gamma$ column shows the integrated results calculated according to Equation (4). As shown, the primary performance improvement stems from mask classifation abillity of MaskCLIP++.
>
> (7) **Response to "original similarity relationships are noisy and require improvement"**:
> We agree that this statement summarizes the goal of most OVS methods. However, the motivations and approaches differ significantly. Our research does not aim to explore how to adapt mask shapes to CLIP or relearn pixel grouping and aggregation relationships. Instead, we focus on enhancing CLIP's mask classification capabilities using limited image segmentation data. The key to avoiding overfitting lies in ensuring that the alignment remains consistent before and after applying additional parameters $\theta$.

---

> > ### Author Response · Authors · 2024-11-22
> >
> > (8) **Response to Concerns Regarding Originality**: Given the strong performance of prior work such as CAT-Seg, we respect and understand why reviewer n1UQ may have approached our work with preconceived concerns regarding originality and fairness. However, we would like to take a step back and compare our work with CAT-Seg from multiple perspectives within the context of our approach. **This comparison aims to highlight our contributions clearly and address any concerns regarding fairness.**:
> >   - **PSMs do Not perform cost aggregation**: The core of CAT-Seg lies in cost aggregation, achieved by designing a complex decoder connected to CLIP and using shallow visual features as priors. This approach learns, on COCO data, which pixel-text pairs are "similar" and which are "different." **In contrast, our motivation stems from the belief that existing mask generators already possess a degree of class-agnostic "pixel aggregation" capability.** Thus, we do not additionally learn how to aggregate pixels but instead focus on adapting CLIP for local feature transfer. This focus is also the foundation of the efficiency of our fine-tuning method.
> >    - **CAT-Seg + Mask Generator does Not perform better:** In our response to Reviewer gVpU under *"Inference with the same masks"*, we proposed a fair inference paradigm: using the exact same mask generator to produce masks, without employing any additional parameters outside of CLIP (such as our PSM or CAT-Seg's decoder), and without integrating classification scores from the mask generator. We simply load CLIP parameters from different sources into the same CLIP architecture. Under this inference paradigm, **our approach produced a better CLIP for mask classification**. One potential explanation, which we validated in the second part of the ablation study (Table 2), is that fine-tuning CLIP for region-level recognition is less challenging than fine-tuning it for pixel-level recognition.
> >    - **Other advantages compared to CAT-Seg**:
> >       - **Functionality**: Mask-based OVS methods can perform instance-level segmentation, while pixel-based OVS methods cannot.
> >       - **Training Efficiency**: Figure 4 illustrates the efficiency in terms of training time and memory usage. In our response to Reviewer mMjr's *"Discussion on data quantity and quality"*, we demonstrated the data efficiency of our fine-tuning approach.
> >       - **Inference Speed**: The following table records the average inference latency on A-150 on the same device (a RTX 3090 GPU), comparing CAT-Seg and our method with large-level VLM as shown in Table 5.
> >
> >       | Method | Latency(ms) |
> >       |:------:|:------:|
> >       |CAT-Seg | 47.2   |
> >       |Ours    | 30.0   |
> >
> > We sincerely thank Reviewer n1UQ for valuable feedback and thoughtful suggestions. We look forward to further discussions. In the revised paper, we will highlight all changes in blue and ensure they are consistent with our responses.

---

> > > ### Comment · Reviewer_n1UQ · 2024-11-25
> > >
> > > > Our innovation lies in the fine-tuning approach, without altering the inference paradigm of mask-based OVS.
> > >
> > > My concern is that this method is based on a mask-based OVS aimed at improving mask accuracy. Therefore, why does it claim that this method does not require a mask generator?
> > >
> > > > Response to Concerns Regarding Originality: Given the strong performance of prior work such as CAT-Seg, we respect and understand why reviewer n1UQ may have approached our work with preconceived concerns regarding originality and fairness. However, we would like to take a step back and compare our work with CAT-Seg from multiple perspectives within the context of our approach.
> > >
> > > In the paper:
> > > > The pretrained CLIP mask embedding Em and the category text embedding $E_t$ are initially pre-aligned with the initial similarity map $S = ⟨E_m, E_t⟩$ being relatively coarse. Optimizing this alignment with some supervision is needed to achieve a more accurate similarity map.
> > >
> > > CAT-Seg aggregates the cost volume (similarity between visual and text, coarse masks) to obtain improved masks. Similarly, this method uses PSM to enhance the similarity between $E_t$ (text) and $E_m$ (visual), resulting in higher mask accuracy. It seems that the core idea behind these two methods appears to be fundamentally similar.

---

> > > > ### Author Response · Authors · 2024-11-27
> > > >
> > > > ## Why we mentioned "no need for a mask generator"
> > > >
> > > > As shown in Figure 2, to the best of our knowledge, previous mask-based OVS methods, whether adapting the mask generator to CLIP or CLIP to the mask generator, have relied on using generated masks as input to a layer of CLIP, with ground truth (GT) masks only as training targets. The goal has been to create a synery between CLIP and the mask generator. However, based on the analysis around Figure 1, we have drawn the following two points:
> > > >
> > > > - From the perspective of mask generation, previous methods that trained mask generators, whether in open or closed vocabularies, still have significant untapped potential in generating masks for new categories (as shown in Figure 1b and Table 3). This suggests that, given the same dataset, spending excessive time training the mask generator may not yield significant benefits.
> > > > - From the perspective of mask classification, the performance gap between GT masks and generated masks indicates that prior mask-based OVS methods' use of generated masks to guide CLIP during training may be bad for alignment (as demonstrated in the ablation study in Table 2). On the other hand, the potential for improvement in Figure 1a is far less than that in Figure 1b, suggesting that CLIP's classification ability urgently needs enhancement.
> > > >
> > > > Therefore, our argument for not needing a mask generator is discussed in the context of  fine-tuning. The previous abstract may have been misunderstood because the sentences were too long. We have revised the abstract.
> > > >
> > > > ## On "the core idea is fundamentally similar":
> > > > (1) Before addressing the statement in Section 3.2 that you quoted, we would like to emphasize the key issues and motivations behind our work.
> > > >
> > > > We chose a mask-based open-vocabulary segmentation (OVS) approach and made the following observations:
> > > >
> > > > - We identified untapped potential in mask generators;
> > > > - We argued that high-quality image segmentation data is crucial for improving CLIP's mask classification capabilities.
> > > >
> > > > Based on these observations, we found that previous methods did not meet our needs:
> > > >
> > > > - Existing work mainly focused on adapting the mask generator and CLIP during fine-tuning;
> > > > - There is a lack of research on how to fine-tune CLIP itself to enhance its mask recognition abilities across various categories, especially when segmentation data is limited in quantity and diversity.
> > > >
> > > > (2) Next, please allow us to provide further clarification on the context of the sentence you quoted.
> > > >
> > > > In Section 3.2, paragraph 2, we first state:
> > > >
> > > > > ... Optimizing this alignment with some supervision is needed to achieve a more accurate similarity map. We refer to this process as parameterized similarity modeling (PSM). ...
> > > >
> > > > This is meant to highlight that PSM is a universal pursuit in OVS tasks, where nearly all works aim to optimize the image-text similarity.
> > > >
> > > > The central discussion that follows is that, when fine-tuning CLIP with very limited data, achieving an alignment without categorical bias is not trivial. In the simplest linear layer modeling, we analyze a failure mode and point out that the root cause of failure is the disruption of consistent alignment.
> > > >
> > > > Subsequently, we provide two modeling approaches that maintain **consistent alignment** based on the original CLIP pooling method. In Table 1 and Table 11, we experimentally show the relationship between the "PSM structure," "consistent alignment," and "overfitting to training categories."
> > > >
> > > > This is the main knowledge we intended to convey in the paragraph, which goes beyond the first few sentences.
> > > >
> > > > (3) As you pointed out, MaskCLIP++ and CAT-Seg differ in their objectives. CAT-Seg optimizes the $HW \times K$ similarity map and directly uses this map as the segmentation result. This requires several additional Swin Transformer blocks to learn the relationships between the $H \times W$ pixels. In contrast, in our approach, given the $Q$ masks, we do not build relationships between these $Q$ masks, but instead, we focus on leveraging the $Q \times K$ similarity and maintaining consistent alignment before and after PSM to improve CLIP’s ability to classify masks.
> > > > The reason we avoid learning pixel-level similarity relationships stems from our analysis in introduction. This distinguishes our motivation from CAT-Seg and allows for lower fine-tuning costs.
> > > >
> > > > (4)
> > > > In Table 7, we further demonstrate that, under the same generated masks, the CLIP fine-tuned with CAT-Seg performs worse in mask classification compared to MaskCLIP++.
> > > >
> > > > | Method | PC-459 | A-150 | PC-59 | Stuff |
> > > > |:------:|:------:|:------:|:------:|:------:|
> > > > |CAT-Seg | 19.6   | 34.1  | 57.3  | 45.6  |
> > > > |MaskCLIP++| 21.3 | 34.8  | 59.3  | 47.2  |
> > > >
> > > > Overall, **we argue that the term 'enhance the similarity' should not be used to encapsulate our core idea, and our contribution deserves more careful consideration.**
> > > > Regardless, we appreciate your valuable suggestions, which have also helped us recognize potential ambiguities in the abstract.

---

> > > > > ### Comment · Reviewer_n1UQ · 2024-11-28
> > > > >
> > > > > > Why we mentioned "no need for a mask generator"
> > > > >
> > > > > Thank you for clarifying this point and addressing my concern.
> > > > >
> > > > > > On "the core idea is fundamentally similar"
> > > > >
> > > > > I appreciate the detailed explanation of the key issues and motivations behind your work. Let us consider the core idea from a high-level perspective. The PSM framework can be abstracted as:
> > > > >
> > > > > $$ S^* = \theta S = \theta<V,T> \approx <\theta V, \theta T>$$
> > > > >
> > > > > where $V$ and $T$ denote the visual and text embeddings respectively. $S$ represents the similarity map, $\theta$ is a learnable module. For simplicity, I have omitted the norm and residual connections, as I believe they do not impact the core idea.
> > > > >
> > > > > As far as I can see, the differences between PSM and CAT-SEG can be summarized as follows:
> > > > > - In PSM, $\theta$ is a Linear layer, whereas in CAT-SEG, $\theta$ is implemented as a Swin Transformer.
> > > > > - In PSM, $V$ represents mask embeddings, while in CAT-SEG, $V$ corresponds to patch embeddings.
> > > > >
> > > > > The similarity between these approaches lies in the application of $\theta$ to both $𝑉$ and $T$, ensuring that **the visual and text representations remain in the same subspace parameterized by $\theta$**. This preserves the original visual-text alignment established by CLIP, which is crucial for maintaining generalization ability. And I believe this mechanism is the key to overcoming the overfitting problem.
> > > > > Thus, while PSM and CAT-SEG have distinct implementations and motivations, I am inclined to consider their core mechanism to be similar.
> > > > >
> > > > > > After mask embeddings are updated via gradient descent, the order of $r_1$ and $r_2$ is closer to the target, but the order of $s_1$ and $s_2$ may deviate further from the target.
> > > > >
> > > > > Thank you for the toy example in the Appendix. While the authors emphasize the concept of consistency alignment, I find myself questioning the necessity of maintaining the order of $s_1$ and $s_2$. Since the final result is derived from $r$, not the intermediate coarse map $s$, the importance of preserving the order in $s$ seems less compelling to me.

---

> > > > > > ### Author Response · Authors · 2024-11-28
> > > > > >
> > > > > > Thank you again for your thoughtful and detailed comments.
> > > > > >
> > > > > > I agree with your analysis of the similarities and differences between consistency alignment and cost aggregation.
> > > > > >
> > > > > > I also understand your concerns regarding the necessity of consistency alignment. For the sake of academic discussion (not rebuttal), we would like to clarify the source of these concerns by outlining our reasoning.
> > > > > >
> > > > > > We illustrate through the toy case in Figure 5 and the statistical results in Table 11 that, when fine-tuning with an 'overfitted PSM,' the similarity before and after applying PSM shows inconsistency, resulting in a significant performance gap when PSM is either removed or retained. This led us to the empirical proposition (1):
> > > > > >
> > > > > > $$
> > > > > > \text{overfitting} \Rightarrow \text{inconsistency alignment}   \quad (1)
> > > > > > $$
> > > > > >
> > > > > > If proposition (1) holds, then proposition (2) should also hold:
> > > > > >
> > > > > > $$
> > > > > > \text{consistency alignment} \Rightarrow \text{not overfitting} \quad (2)
> > > > > > $$
> > > > > >
> > > > > > Therefore, we believe that the design of PSM should consider this consistency alignment constraint.
> > > > > >
> > > > > > However, since the truth of proposition (1) is based on experimental observation, proposition (2) cannot be strictly proven. This may be why it seems "less compelling".
> > > > > >
> > > > > > We believe that identifying a more precise and implementable factor $X$, such that $X \Rightarrow \text{not overfitting}$, is a direction still worth exploring further in the future.

---

> ### Author Response · Authors · 2024-11-25
>
> Dear Reviewer,
>
> Thank you once again for your valuable feedback and comments on our manuscript. We have made revisions based on your suggestions and addressed the concerns raised. We kindly ask if you could confirm whether our responses have sufficiently addressed your concerns.
>
> We appreciate your time and effort in reviewing our work.

---

### Official Review · Reviewer_gVpU · 2024-10-30

**Soundness:** 2
**Presentation:** 3
**Contribution:** 2
**Rating:** 5
**Confidence:** 4

**Summary:**

This paper propose MaskCLIP++ to transfer CLIP's image-level recognition to local regions via ground truth masks and labels. Based on the analysis of the sources of misalignment, this paper proposes a consistency alignment constraint to ensure that alignment optimization preserves CLIP’s original embedding space. The proposed MaskCLIP++ can be used with different mask generators to achieve open-vocabulary segmentation at the semantic- or instance-level and achieve superior performance.

**Strengths:**

(1) The proposed MaskCLIP++ is straightforward and easy to follow.

(2) The presentation of this paper is good, with outstanding writing style and clarity.

(3) The paper demonstrates obvious improvements compared to the original CLIP, and it performs well with various
baseline segmentation methods.

(4) Detailed ablation studies are performed to analyze the impact of each component.

**Weaknesses:**

**[About Methods]**

(1) The illustration of PSM is confusing and unclear. For example, in *Line 204-210*, the author analyzes the cause of misalignment and offers an unintuitive phenomenon. Since the similarity $r$ can be optimized to be close to the target, why the order of $m'^T_1t$ may deviate further from the target? It would be better if the author could explain this phenomenon using specific statistics or figures.

(2) In *Line 236-240*, the author argues that the simplest way to achieve consistency alignment is to update the similarity map in another dimension. If I understand right, it refers to $Linear<E_m, E_t>$ as shown in Table 1. However, the next paragraph directly demonstrates $<E_m, Norm(E_t+P_t)>$, which is incoherent and irrelevant to the above statement. Besides, the $<E_m, Norm(E_t+P_t)>$ used in this paper is very similar to the Content-Dependent Transfer module in [1], resulting in the limited contribution of this paper.

(3) I doubt the effectiveness of decoupling the joint training of the mask generator and classifier for open-vocabulary segmentation. It may be challenging to achieve "real open vocabulary" without training the mask generator. For example, if I want to perform part segmentation (e.g., segmenting person's hands rather than person), the off-the-shelf mask generator may struggle to generate these fine-grain masks if not training on the target dataset.

**[About Experiments]**

(1) As far as I understand, MaskCLIP++ is an enhancement of CLIP to align vision and language in regional representations. To validate the local classification ability, it should be compared to related counterparts like MaskCLIP [2], CLIP-Surgrey [3], SCLIP [4], CLIPSelf[ 5], SAM-CLIP [6], etc. A more valid usage is to perform pixel-level classification based on dense features like SCLIP (zero-shot segmentation), which avoids the impact of mask generators and only focuses on local representations.

(2) It also makes sense to combine MaskCLIP++ with existing open-vocabulary segmentation or mask generators to improve their classification performance as done in this paper. However, some comparative experiments should be conducted using counterpart methods mentioned above (e.g., FC-CLIP to generate mask and SAM-CLIP to classify).

(3) The scenes and categories in ADE and COCO Panoptic are similar and relevant, which is not convincing to assess overfitting. More scenes like Street-View Datasets used in FC-CLIP should be adopted for better persuasiveness.

---

[1] Collaborative Vision-Text Representation Optimizing for Open-Vocabulary Segmentation.

[2] Extract Free Dense Labels From CLIP.

[3] CLIP Surgery for Better Explainability with Enhancement in Open-Vocabulary Tasks.

[4] SCLIP: Rethinking Self-Attention for Dense Vision-Language Inference.

[5] CLIPSelf: Vision Transformer Distills Itself for Open-Vocabulary Dense Prediction.

[6] SAM-CLIP: Merging Vision Foundation Models towards Semantic and Spatial Understanding.

**Questions:**

(1) What is the detailed setting used in Figure 1b? How to determine the GT classification of a mask if it contains multiple categories?

(2) Why use the OpenAI model for ResNet architecture but EVA-CLIP for ViT architecture?

(3) Section 4.1 demonstrates that the model is fine-tuned on the COCO Panoptic training set, but in Table 5, the training set is COCO-Stuff. Why use different training sets?

---

> ### Author Response · Authors · 2024-11-22
>
> Thank you to Reviewer gVpU for acknowledging the simplicity and effectiveness of our method, as well as the presentation of the paper. We will provide further clarification on the concerns raised by gVpU.
>
> **[About Methods]**
>
> (1) **Regarding the design principles of PSM**: We will include the following experiments to verify whether PSM significantly violates the consistency alignment constraints and to explain the relationship between maintaining consistency constraints and overfitting. After fine-tuning with a specific PSM, we will compare performance when PSM is retained or removed during inference. As shown in the table below, **PSM designs that significantly violate consistency alignment lead to better performance when PSM is removed during inference, but perform much worse on unseen categories after PSM application compared to the baseline**. In contrast, PSM designs that largely maintain consistency alignment show consistent improvements over the baseline, both in the presence or absence of PSM, with gains observed in both seen and unseen categories.
>
> |                         PSM                         | Use in test | mIoU(S) | mIoU(U) | mIoU |
> | :-------------------------------------------------: | :---------: | :-----: | :-----: | :--: |
> |                          -                          |      -      |  32.7   |  16.3   | 23.3 |
> |  $\langle \operatorname{Linear}(E_m), E_t \rangle$  |      N      |  37.7   |  20.2   | 27.7 |
> |                                                     |      Y      |  32.4   |   1.9   | 14.9 |
> |  $\langle E_m, \operatorname{Linear}(E_t) \rangle$  |      N      |  37.6   |  20.3   | 27.6 |
> |                                                     |      Y      |  32.5   |   1.8   | 14.9 |
> |   $\operatorname{Linear}\langle E_m, E_t \rangle$   |      N      |  45.2   |  24.6   | 33.4 |
> |                                                     |      Y      |  45.4   |  25.7   | 34.1 |
> |   $\langle E_m, \operatorname{Norm}(P_t) \rangle$   |      N      |  39.5   |  21.4   | 29.1 |
> |                                                     |      Y      |  31.9   |   4.5   | 16.2 |
> |   $\langle E_m, \operatorname{Norm}(E_t) \rangle$   |      N      |  45.8   |  24.4   | 33.5 |
> |                                                     |      Y      |  46.2   |  25.2   | 34.1 |
> | $\langle E_m, \operatorname{Norm}(E_t+P_t) \rangle$ |      N      |  44.0   |  24.1   | 32.6 |
> |                                                     |      Y      |  46.3   |  26.7   | 35.1 |
>
>
>
> (2) **Regarding the similarity with the CDT module**: Cross-attention structures similar to Content-Dependent Transfer (CDT) are widely used in multimodal alignment. However, we aim to highlight that, **when fine-tuning with limited data, the key to maintaining consistency alignment lies in the $\langle E_m, \operatorname{Norm}(E_t) \rangle$ part**, whereas $\langle E_m, \operatorname{Norm}(P_t)\rangle$ does not utilize the pre-alignment between $E_m$ and $E_t$. **We aim to emphasize the importance of the consistency alignment design principle through Table 1, rather than proposing a specific architecture for the PSM.** As for the gains brought by $P_t$ in $\langle E_m, \operatorname{Norm}(E_t+E_p) \rangle$, as discussed in Appendix Table 13, they primarily supplement the global visual information missing from $E_m$.
>
> (3) **Regarding multi-granularity open-vocabulary segmentation**: To the best of our knowledge, existing mask-based OVS methods train mask generators  on the COCO dataset. While their mask generation is class-agnostic to some extent, the extent to which they can achieve multi-granularity segmentation masks remains an open research question. **Our work primarily focuses on enhancing CLIP's mask classification capability without making improvements to the mask generator.**

---

> > ### Author Response · Authors · 2024-11-22
> >
> > **[About Experiments]**
> >
> > (1) **Inference without masks:**
> > The optimization goal of our method is to enhance CLIP's mask classification ability, rather than improving its pixel-level classification.
> > In response to Reviewer gVpU's request, we combine our MaskCLIP++ with various pixel-level inference methods.
> > As shown in the table below, our MaskCLIP++ improves performance on top of these methods. (All methods use OpenAI-ViT-B/16 CLIP).
> >
> >
> > | Inference method    |A-847|PC-459|A-150| PC-59|COCO-Stuff|Cityscapes|
> > |:---:|:---:|:---:|:---:|:---:|:---:|:---:|
> > | SCLIP               | - | 7.7 | 16.1 | 34.2 | 22.4 | 32.2 |
> > | SCLIP w/ Ours       | - | 9.5 | 19.3 | 40.8 | 29.0 | 35.9 |
> > | ClearCLIP           | - | 7.9 | 16.7 | 36.0 | 23.9 | 30.0 |
> > | ClearCLIP w/ Ours   | - | 9.6 | 19.2 | 40.8 | 30.4 | 33.5 |
> > | ProxyCLIP           | 7.4 | 8.4 | 20.2 | 39.1 | 26.5 | 38.1 |
> > | ProxyCLIP w/ Ours   | 8.0 | 9.9 | 23.2 | 44.6 | 34.2 | 42.4 |
> >
> > (2) **Inference with the same masks:** Since SAM-CLIP has not been open-sourced, we extracted the CLIP parameters from two competitive open-source methods (CLIPSelf and CAT-Seg) and performed mask-based inference using the same generated masks. As we do not train a PSM on their CLIP, the results in the table below are obtained **without using the PSM and the ensemble strategy**.
> >
> >
> > | CLIP arch. | CLIP params | PC-459 | A-150 | PC-59 | COCO-Stuff |
> > |:----------:|:---:|:-----:|:---:|:---:|:---:|
> > |OpenAI-CLIP-ViT-L|Origin  |9.3  |17.6 |27.4 |18.0 |
> > |                 |CAT-Seg |19.6 |34.1 |57.3 |45.6 |
> > |                 |Ours    |21.3 |34.8 |59.3 |47.2 |
> > |EVA-CLIP-ViT-L/14|Origin  |11.1 |20.1 |26.0 |18.3 |
> > |                 |CLIPSelf|13.0 |22.9 |39.1 |27.1 |
> > |                 |Ours    |24.2 |37.8 |62.3 |50.6 |
> >
> >
> >
> > (3) **Results on street-view datasets:** We add the open-vocabulary semantic segmentation performance on Cityscapes and BDD100K. Using the same generated masks and CLIP-ConvNeXt-L as in FC-CLIP, the results are shown in the table below.
> >
> > | Method | Cityscapes | BDD100K |
> > |:------:|:------:|:------:|
> > |FC-CLIP | 56.2   | 53.8   |
> > |MaskCLIP++| 56.3 | 55.9   |
> >
> > ---
> >
> > **[Questions]**
> >
> > (1) **Setting of Fig. 1b:** The statistics in Fig. 1b primarily aim to evaluate the potential for improving mask classification. For the same mask generator:
> >
> >    - **CLIP classification** represents the performance level achievable using CLIP for classification.
> >    - **GT classification** assumes *perfect classification* of the generated masks.
> >
> >    To achieve this *perfect classification* effect, we measure IoU and use the Hungarian algorithm to assign predicted masks to ground truth (GT) masks, thereby obtaining the perfect category labels for the predicted masks. Merging these GT masks produces the performance represented by the blue dashed line in Fig. 1b.
> >
> > (2) **Source of pretrained CLIP:** As shown in the table under "Inference with the same masks," EVA-CLIP demonstrates better initialization performance compared to OpenAI-CLIP, both before and after fine-tuning. Therefore, we selected EVA-CLIP for the ViT architecture.
> >
> > (3) **Choice of Fine-tuning datasets for OVSS and OVPS:** This choice was made to align with the settings used by most methods in Tables 5 and 6. Specifically, COCO-Stuff is used for fine-tuning when only reporting OVSS performance, while COCO-Panoptic is used when reporting both OVPS and OVIS performance. We correspondingly use mask generators trained on COCO-Panoptic and COCO-Stuff.
> >
> > We sincerely thank Reviewer gVpU for valuable feedback and thoughtful suggestions. We look forward to further discussions. In the revised paper, we will highlight all changes in blue and ensure they are fully aligned with our responses.

---

> > > ### Comment · Reviewer_gVpU · 2024-11-25
> > >
> > > Thanks for the authors’ responses, which have resolved some of my concerns. However, some answers seems deviate my original intention. Specifically,
> > >
> > > > (1) The illustration of PSM is confusing and unclear. For example, in *Line 204-210*, the author analyzes the cause of misalignment and offers an unintuitive phenomenon. Since the similarity $r$ can be optimized to be close to the target, why the order of $m'^T_1t$ may deviate further from the target? It would be better if the author could explain this phenomenon using specific statistics or figures.
> > >
> > > While some results of consistency alignment are presented in Table 1, I still remain unclear about the formation and underlying causes of this unintuitive phenomenon. For example,  when analyzing the cause of misalignment, why `after mask embeddings are updated to m′ via gradient descent, the order of` $m^{′T}_1t$ `and` $m^{′T}_2t$ `may deviate further from the target.` It would be helpful to provide a clear demonstration using specific images to illustrate the similarity deviation of $m_1$ and $m_2$.
> > >
> > > > (2) In *Line 236-240*, the author argues that the simplest way to achieve consistency alignment is to update the similarity map in another dimension. If I understand right, it refers to $Linear<E_m, E_t>$ as shown in Table 1. However, the next paragraph directly demonstrates $<E_m, Norm(E_t+P_t)>$, which is incoherent and irrelevant to the above statement. Besides, the $<E_m, Norm(E_t+P_t)>$ used in this paper is very similar to the Content-Dependent Transfer module in [1], resulting in the limited contribution of this paper.
> > >
> > > The authors claim that this paper does not focus on proposing a specific architecture for the PSM, but it does adopt the advanced  $<E_m, Norm(E_t+P_t)>$ rather than the naive consistency alignment design $Linear<E_m, E_t>$. As mentioned in the question above, the introduction of $<E_m, Norm(E_t+P_t)>$ is abrupt, resulting in a lack of coherence between the description in Lines 238-242 and the subsequent paragraph. Additionally, it appears that the improvement of MaskCLIP++ partially relies on this "unfocused" specific architecture $<E_m, Norm(E_t+P_t)>$, making it less convincing to validate the effectiveness of the consistency alignment design. Would MaskCLIP++ still outperform other works based on the simpler $Linear<E_m, E_t>$?
> > >
> > >
> > > > (3) I doubt the effectiveness of decoupling the joint training of the mask generator and classifier for open-vocabulary segmentation. It may be challenging to achieve "real open vocabulary" without training the mask generator. For example, if I want to perform part segmentation (e.g., segmenting person's hands rather than person), the off-the-shelf mask generator may struggle to generate these fine-grain masks if not training on the target dataset.
> > >
> > > I acknowledge that existing mask-based OVS methods face limitations in addressing multi-granularity open-vocabulary segmentation.  But MaskCLIP++ seems less adaptive than existing OVS methods in certain cases. For example, when we want to perform part segmentation, previous works can jointly train the mask generator and classifier with image-text data, while MaskCLIP++ needs to train both the mask generator and classifier separately using pixel-level data? Given that this work primarily focuses on improving CLIP’s mask classification capability, would it be more appropriate for the title and main paper to emphasize classification rather than segmentation?
> > >
> > > > Source of pretrained CLIP: As shown in the table under "Inference with the same masks," EVA-CLIP demonstrates better initialization performance compared to OpenAI-CLIP, both before and after fine-tuning. Therefore, we selected EVA-CLIP for the ViT architecture.
> > >
> > > I understand and agree with the decision to use EVA-CLIP, but I am curious whether other baseline methods also employed this powerful version. Ensuring consistency in this regard is important for a fair comparison.

---

> ### Author Response · Authors · 2024-11-25
>
> Dear Reviewer,
>
> Thank you once again for your valuable feedback and comments on our manuscript. We have made revisions based on your suggestions and addressed the concerns raised. We kindly ask if you could confirm whether our responses have sufficiently addressed your concerns.
>
> We appreciate your time and effort in reviewing our work.

---

> ### Author Response · Authors · 2024-11-25
>
> We sincerely apologize for not fully addressing your concerns in our previous response. Thank you for your patience and for providing a more detailed explanation of your doubts. Allow us to provide additional clarifications in a structured manner:
>
> (1) We add an explanation and **Figure 5 in the second paragraph of Appendix C.1** to clarify the phenomenon of the similarity deviation. Table 11, presented thereafter, statistically demonstrates the prevalence of this phenomenon across different PSMs.
>
> (2) As stated in the first paragraph of Section 4.2:
> > when fine-tuning CLIP models with attention pooling (ViT and ResNet), we still use the PSM of $\operatorname{Linear}\langle E_m, E_t \rangle$.
>
>   The Large VLMs presented in Table 5 and Table 6 are based on the ViT architecture, which demonstrates that MaskCLIP++ still outperforms other methods based on the simpler $\operatorname{Linear}\langle E_m, E_t\rangle$.
>
> (3) To explore how our method performs on partial segmentation tasks in open-vocabulary settings, we present the following insights:
>
> - **Fine-tuning MaskCLIP++ on part-level segmentation data is not required**. The following experiment demonstrates this: using groundtruth masks from the same **MHP-v1[1] dataset**, MaskCLIP++ **fine-tuned on COCO-Stuff** still achieves a significant improvement in mask classification capability on this dataset. (The 19 categories in MHP-v1, such as left leg, face, and sunglasses, are all novel to categories in COCO-Stuff.)
>
>   |      | mIoU |
>   |:----:|:----:|
>   |Origin CLIP| 30.6 |
>   |MaskCLIP++ | 48.6 |
> - **For the mask generator, it may be necessary to retrain it on masks with a similar granularity.** Our findings regarding the potential of the existing mask generator are as follows: if the model has seen an example like an elephant during training, it can still generate masks for a **similar-granularity object** like a giraffe. However, it is known that mask generators, such as Mask2Former, trained at the semantic level typically perform poorly for instance-level segmentation. Therefore, it is hypothesized that they would also struggle to generate effective mask proposals at the part level. In addition to retraining the mask generator, more unified mask generators, such as OneFormer[2], may be worth exploring in the future.
>
> Following your suggestion, we have revised the abstract to emphasize the improvements we made in mask classification.
>
> (4) To the best of our knowledge, previous works using EVA-CLIP include CLIPSelf and Cat-Seg. In our initial response (which has been added to Table 7), we compare the mask classification ability of the CLIPs from MaskCLIP++, CLIPSelf, and Cat-Seg. In addition, CAT-Seg provides their performance using EVA-CLIP in the appendix of their paper. Under the Cat-Seg method, there does not appear to be a clear advantage of EVA-CLIP over OpenAI's CLIP.
>
>
> We hope that the revisions and responses provided have addressed your concerns clearly. Please let us know if any further clarification is needed.
>
> ---
>
> [1] Li, Jianshu, et al. "Multiple-human parsing in the wild." arXiv preprint arXiv:1705.07206 (2017).
>
> [2] Jain, Jitesh, et al. "Oneformer: One transformer to rule universal image segmentation." Proceedings of the IEEE/CVF Conference on Computer Vision and Pattern Recognition. 2023.

---

> > ### Comment · Reviewer_gVpU · 2024-11-26
> >
> > Thank you for the explanation. However, there appear to be some inconsistencies between Figure 5 and the Method section. For instance, in Line 207, the authors state that `the mask embeddings are updated using a parameter Θ,` but in Line 208 and Figure 5, Θ is used for updating the text embedding. Then, in Line 210,  𝑚′ is described as the updated mask embedding again. This contradiction makes it difficult to follow and understand the process.
> >
> > Moreover, the authors have not yet clarified the rationale behind the sudden introduction of $<E_m, Norm(E_t+P_t)>$ in the method section.  In my view, the proposed PSM $Linear<E_m, E_t>$, which is the core of this paper, seems to overlap with CAT-Seg (cost volume embedding), while the $<E_m, Norm(E_t+P_t)>$ is a simple mimicry of  Content-Dependent Transfer used in MAFT+, without introducing any significantly novel techniques.

---

> ### Author Response · Authors · 2024-11-27
>
> (1) First, we appreciate your pointing out the inconsistencies between the Figure 5 and the main text. We have made the necessary revisions, using $\Theta$ updating the mask embeddings as an example.
>
> (2) We did not provide extensive background for introducing $\langle E_m, \operatorname{Norm}(E_t + P_t) \rangle$ because we are not proposing a specific PSM structure (we also referenced MAFT+ when introducing it). Instead, it arises from our understanding of PSM design. Clearly, there are multiple reasonable PSM structures, but they should all adhere to the consistency alignment constraint as closely as possible.
>
> (3) For the clarification of core idea of our paper, and the differences between our work and CAT-Seg in terms of motivation, explanation, objectives, and effects, please refer to our response to reviewer n1UQ.
>
> We sincerely appreciate your patient and thorough review.

---

> > ### Comment · Reviewer_gVpU · 2024-11-27
> >
> > Thanks for the explanation. After reading the authors’ responses and feedback from other reviewers, I feel that the presentation of the core idea in this paper is somewhat irrational. While the main objective of this paper is to enhance CLIP's mask classification ability through fine-tuning, it appears to overemphasize the role of open-vocabulary segmentation (OVS), making it difficult to understand the underlying rationale between the mask generator and CLIP classifier.
> >
> > First, as a universal mask classification approach, MaskCLIP++ should be capable of classifying masks from multiple sources, such as GT masks, SAM-generated masks, or model predictions. Open-vocabulary segmentation is merely one potential application of this classification method when combined with a mask generator. However, the paper heavily focuses on OVS throughout its method description and evaluation, which not only creates confusion but also fails to comprehensively demonstrate the classification performance. For example, a more convincing way to validate the classification enhancement would be to directly classify GT masks and use a classification-based metric such as **Accuracy**, rather than relying on segmentation-based metrics.
> >
> > Second, this paper claims that the mask generator and CLIP operate as two independent modules in this framework. However, the operation in equation 4 violates this independent principle, making the input to CLIP and the final results (e.g., Table 5) depend on the logits predicted by the mask generator. To maintain independence and generalizability, the fine-tuned CLIP should take class-agnostic masks as input, treating the mask generator as a black box. If logits are used as additional priors, this approach should be clearly distinguished from the original MaskCLIP++.
> >
> > Finally, the inconsistent presentation of the PSM in the original method section and the rebuttal stage (e.g., Figure 5) significantly hinders the readability of this paper. I recommend that the authors carefully check the manuscript to ensure consistency and avoid such confusion in other parts of the paper.
> >
> > In summary, as a mask classification method, I believe MaskCLIP++ should not revolve solely around OVS. The paper would benefit from a reorganization that focuses on the classification roadmap. To evaluate the classification improvement, classification-based metrics should be employed, and comparisons with relevant works should be made. To demonstrate its generality, the method should be tested on multi-domain and multi-granularity evaluation sets. Once the classification capability is well demonstrated, the authors could explore its application to OVS by leveraging existing mask generators, as done in this work. Note that I am not requesting the authors to conduct these experiments during the discussion, but rather suggesting a potential improvement direction.
> >
> > Overall, I believe this paper requires major revisions to improve its presentation and clarity. Additionally, I am somewhat dissatisfied that some of my earlier concerns appear to have been overlooked during rebuttal, which compels me to reiterate them. As a result, I will maintain my original score.

---

> > > ### Author Response · Authors · 2024-11-29
> > >
> > > Thank you for your comments.
> > >
> > > 1. I agree that mask accuracy is a more direct evaluation metric and could be included as supplementary information in the future. **However, under the condition of using the same generated masks, the segmentation metrics also reflect improvements in mask segmentation performance.** Compared to mask accuracy, we chose segmentation metrics for the following reasons:
> > >    - In the introduction, we need to clarify why we focus on improving mask classification rather than mask generation. mIoU is a metric that reflects both mask classification and mask generation performance. We believe that using this metric to compare the potential improvements of both is the most intuitive. Additionally, continuing to use this metric in subsequent experiments ensures consistency with the introduction.
> > >    - Researchers in the OVS field are more familiar with segmentation metrics like mIoU.
> > >
> > > 2. In our understanding, tables like Table 5 that compare with state-of-the-art (SOTA) methods typically present overall performance, rather than isolating the performance gains attributed to the core technique alone. A fair comparison of the impact of various techniques on performance is usually provided in the ablation study. For example, the importance of consistency alignment is shown in Table 1; the quality of priors in Table 2; the impact of the mask generator and ensemble strategy in Table 3; the effect of different CLIP architectures in Table 4; the comparison with prior work on fine-tuning CLIP for mask classification in Table 7; and the combination with various methods that do not use a mask generator in Table 8.
> > >
> > >    Regarding the ensemble issue, first, its performance improvement is marginal for us (Table 3, Response to LpvX). Secondly, many methods in Table 5 also use ensemble strategy (ZSSeg, ODISE, FC-CLIP, ...). These papers report their performance after ensemble in their SOTA comparison tables as well.
> > >
> > > 3. We appreciate your suggestion and will carefully check whether there are any other inconsistencies in the main text after the temporary addition of Figure 5.
> > >
> > > 4. We also accept the suggestion to conduct more extensive evaluations on multi-domain and multi-granularity datasets in the future, and compare with relevant methods using the mask classification metric.
> > >
> > > 5. We apologize for having caused you to reiterate some concerns.
> > >
> > >
> > > In our recent discussions, we have learned a great deal while continuously working to address your concerns. Although we may not end up agreeing on everything, we sincerely appreciate your thorough and thoughtful feedback.

---

### Official Review · Reviewer_mMjr · 2024-11-04

**Soundness:** 3
**Presentation:** 3
**Contribution:** 3
**Rating:** 6
**Confidence:** 4

**Summary:**

The paper addresses limitations in open-vocabulary image segmentation, particularly the reliance on low-quality masks that weaken vision-language alignment. The authors propose MaskCLIP++, a fine-tuning framework that transfers CLIP's image-level recognition to local regions using ground truth masks and labels, eliminating the need for a specific mask generator. They introduce a consistency alignment module during fine-tuning to enhance local image-text similarity and reduce overfitting. Empirical results show that MaskCLIP++ outperforms previous state-of-the-art mask-based segmentation methods, achieving improvements of +1.7, +2.3, +2.1, +3.1, and +0.3 mIoU on the A-847, PC-459, A-150, PC-59, and PAS-20 datasets, respectively, with lower training costs.

**Strengths:**

+ The paper provides a good analysis of the limitations of using generated masks in previous open-vocabulary segmentation works, highlighting the challenges in improving mask generation compared to mask classification.

+ The paper introduces a new fine-tuning framework that leverages ground truth masks and labels, eliminating the need for specific mask generators. This approach addresses the limitations of relying on low-quality masks, improving the alignment of vision and language in regional representations.

+ Based on experiments on several benchmarks, the proposed method demonstrates better perfomrances compared to existing state-of-the-art mask-based open vocabulary segmentation methods.

**Weaknesses:**

- While the use of ground truth masks improves performance, it may limit the applicability of MaskCLIP++ in scenarios where such masks are not readily available or are costly to obtain. This may result in limitations of scaling up with data and unfair comparisons with other methods.

- It is not clear whether the insights and proposed method can be generalized/applied to other recent state-of-the-art methods rather than just MaskCLIP. Such experiments/studies could help justify generalization ability of the proposed method.

**Questions:**

Please refer to details of above "Weaknesses" sections. Specifically:

(1) Is that possible to discuss potential ways to address scenarios where ground truth masks may not be available? If there are any techniques you could propose to generate high-quality pseudo-masks when ground truth is unavailable, or if you have explored using a mix of ground truth and generated masks during training.

(2) Ablation studies about applying the proposed method to 1-2 other recent SOTA open-vocabulary segmentation methods will be helpful. This would provide concrete evidence of the method's generalizability beyond just MaskCLIP.

---

> ### Author Response · Authors · 2024-11-22
>
> Thank you to Reviewer mMjr for acknowledging the motivation and contributions of our work. We will provide further clarification on the concerns raised by mMjr.
>
> (1) **Clarification regarding the relationship with MaskCLIP**: We apologize for any confusion caused by the method name "MaskCLIP++." Three recent works have used the name "MaskCLIP" [1, 2, 3], but MaskCLIP++ is not an improved version of any of these works. Since our work focuses on obtaining a CLIP model with improved mask classification capabilities, we named our method MaskCLIP++.
>
> (2) **Regarding fairness**: As described in related work, there are four types of OVS implementations based on data sources. Our method uses a setup with limited image segmentation data (image, masks, and category annotations). The methods compared in Table 5 and Table 6 are all under this setup.
>
> (3) **The combination with SOTA**: To the best of our knowledge, all mask-based OVS methods follow a similar "inference" paradigm: mask generation + mask classification. We focus on improving mask classification rather than mask generation. Therefore, after obtaining fine-tuned CLIP through different fine-tuning strategies, our inference pipeline remains largely consistent with recent works such as FC-CLIP and MAFT+. The mask generators used in Table 5 and Table 6 are derived from the publicly available models of MAFT+ and FC-CLIP.
>
> (4) **Discussion on data quantity and quality**: We understand the reviewer’s concern: “The masks used during inference are often of lower quality compared to the GT masks used during training, and obtaining large-scale GT mask annotations for fine-tuning can be challenging.” We address this concern step-by-step as follows:
>
> - **Impact of generated masks**: Fine-tuning with the generated masks from FC-CLIP (selecting those with the highest IoU to the GT masks) results in a noticeable performance drop, as shown in Table 2. **This indicates that high-quality masks are indeed crucial for fine-tuning**.
> - **data efficiency**: We randomly sampled subsets of COCO at different scales (10%, 1%, and 0.1%) as the training set for fine-tuning. The results, presented in the table below, demonstrate that **enabling pre-trained VLMs to learn localized recognition does not require extensive image segmentation annotations**. This is because we do not rely on segmentation annotations to learn pixel grouping or category information beyond CLIP’s capabilities.
>
> | data size | A-847 | PC-459 | A-150 | PC-59 | PAS-20 | COCO-Stuff |
> | :-------: | :---: | :----: | :---: | :---: | :----: | :--------: |
> |   100%    | 14.5  |  18.7  | 35.4  | 59.1  |  95.8  |    47.1    |
> |    10%    | 14.2  |  19.0  | 34.9  | 59.2  |  95.8  |    47.2    |
> |    1%     | 13.8  |  19.1  | 35.1  | 58.5  |  95.6  |    47.0    |
> |   0.1%    | 12.8  |  18.6  | 34.1  | 58.2  |  94.9  |    46.1    |
>
> In summary, we believe that **for our fine-tuning approach, small but precise annotations are more effective than large-scale noisy annotations**. Data scalability efforts should be directed toward the pre-training of CLIP or mask generators rather than fine-tuning CLIP.
>
>
> We sincerely thank Reviewer mMjr for valuable feedback and thoughtful suggestions. We look forward to further discussions. In the revised paper, all changes will be highlighted in blue and carefully aligned with our responses.
>
> ---
>
> [1] Zhou, Chong, Chen Change Loy, and Bo Dai. "Extract free dense labels from clip." ECCV. 2022.
>
> [2] Ding, Zheng, Jieke Wang, and Zhuowen Tu. "Open-vocabulary universal image segmentation with maskclip." ICML. 2023.
>
> [3] Dong, Xiaoyi, et al. "Maskclip: Masked self-distillation advances contrastive language-image pretraining." CVPR. 2023.

---

### Meta-Review · Area_Chair_MR2w · 2024-12-21

**Metareview:**

In this paper, the authors presented a mask-based CLIP fine-tuning framework for open-vocabulary image segmentation. Specifically, based on the observation that relying on generated low-quality masks could weaken the alignment of vision and language, the authors proposed a new fine-tuning framework (termed MaskCLIP++) using ground-truth masks to enhance CLIP. Experimental results on benchmark datasets show the effectiveness of the proposed method over previous mask-based methods. The strengths of this paper are:
- A good analysis of the limitations of previous methods using generated masks was presented, highlighting the challenges and could be useful for following research in this direction.
- The key idea of MaskCLIP++ is straightforward, simple, and easy to follow.
- An extensive experimental analysis with ablation studies was presented, showing the effectiveness of the proposed method and the contribution of each component.
- The paper is generally well-written and well-organised.

The main weaknesses of this paper include:
- Novelty of the proposed method. There is a major concern around the high similarity of the main contribution of this paper (the consistency alignment PSM) and the cost volume aggregation proposed in prior work CAT-Seg. The rationale behind is also questionable.
- Some technical designs lack details and convincing motivation, e.g. the $<E_m, Norm(E_t+P_t)>$ as mentioned by reviewer gVpU, and other missing/unclear descriptions as mentioned by reviewers n1UQ and LpvX.
- The performance, when compared to other methods, was shown to be insignificant and inferior in some cases.
- Concerns about generalisation and decoupling the joint training of mask generator and classifier.

Overall, although this paper is well-written and shows detailed experiments, there are some major concerns (e.g. novelty and performance) that were not well addressed and need a major revision followed by another round of review. During the second phase of discussions, two reviewers further raised their concerns about this paper (please see section below). Considering all the above, the paper in its current form is not ready to be presented at ICLR, but the AC highly recommend the authors to carefully consider the comments from all the reviewers and revise the paper for a future submission.
The AC understand that this decision might be disappointing and the authors may not agree with it, but hope the above points and the reviewers' comments could be helpful in improving their paper.

**Additional Comments On Reviewer Discussion:**

This paper received review comments from four expert reviewers. During the rebuttal period, there was a heated back-and-force discussion between the authors and reviewers. After the discussion, some of the minor concerns were well addressed by the authors, while there were still some major concerns remaining. In the second phase, i.e. the AC-reviewers discussion, reviewers gVpU and n1UQ further summarised and confirmed their remaining concerns about this paper, while acknowledging the simplicity and (potentially) effectiveness of the method. Their main concerns were summarised in the above section. Through the second phase discussion, both the AC and reviewers agreed on the remained major concerns and a major revision is needed to address them, followed by another round of assessment. The final decision was made mainly according to this, together with the consideration of the quality and completeness of the work to be presented at ICLR.

---

### Decision · Program_Chairs · 2025-01-22

Reject